

**Online measurements of cycloalkanes based on NO[+] chemical**
**ionization in proton transfer reaction time of flight mass**
**spectrometry (PTR-ToF-MS)**
Yubin Chen[1,2], Bin Yuan[1,2,*], Chaomin Wang[3], Sihang Wang[1,2], Xianjun He[1,2],
Caihong Wu[1,2], Xin Song[1,2], Yibo Huangfu[1,2], Xiao-Bing Li[1,2], Yijia Liao[1,2], Min
Shao[1,2]
[1] Institute for Environmental and Climate Research, Jinan University, Guangzhou
511443, China
[2] Guangdong-Hongkong-Macau Joint Laboratory of Collaborative Innovation for
Environmental Quality, Guangzhou 511443, China
[3] School of Tourism and Culture, Guangdong Eco-engineering Polytechnic,
Guangzhou 510520, China
*Correspondence to: Bin Yuan (byuan@jnu.edu.cn)



**Abstract:**

Cycloalkanes are important trace hydrocarbons existing in the atmosphere, and they are considered as a major class of intermediate volatile organic compounds (IVOCs). Laboratory experiments showed that the yields of secondary organic aerosols (SOA) from oxidation of cycloalkanes are relatively higher than acyclic alkanes with the same carbon number. However, measurements of cycloalkanes in the atmosphere are still challenging at present. In this study, we show that online measurements of cycloalkanes can be achieved using proton transfer reaction time-of-flight mass spectrometry with $NO^+$ chemical ionization ($NO^+$ PTR-ToF-MS). Cyclic and bicyclic alkanes are ionized with $NO^+$ via hydride ion transfer leading to major product ions of $C_nH_{2n-1}^+$ and $C_nH_{2n-3}^+$, respectively. As isomers of cycloalkanes, alkenes undergoes association reactions with major product ions of $C_nH_{2n} \cdot (NO)^+$, and concentrations of 1-alkenes and trans-2-alkenes in the atmosphere are usually significantly lower than cycloalkanes (about 25% and <5%, respectively), as the result inducing little interference to cycloalkanes detection in the atmosphere. Calibration of various cycloalkanes show similar sensitivities, associated with small humidity dependence. Appling this method, cycloalkanes were successfully measured at an urban site in southern China and a chassis dynamometer study for vehicular emissions. Concentrations of both cyclic and bicyclic alkanes are significant in urban air and vehicular emissions, with comparable cyclic alkanes/acyclic alkanes ratios between urban air and gasoline vehicles. These results demonstrates that $NO^+$ PTR-ToF-MS provides a new complementary approach for fast characterization of cycloalkanes in both ambient air and emission sources, which can be helpful to fill the gap in understanding importance of cycloalkanes in the atmosphere.



## 1    Introduction

Organic compounds, as important trace components in the atmosphere, are released to the atmosphere from many different natural and anthropogenic sources, which have complicated and diverse chemical compositions (de Gouw, 2005; Goldstein and Galbally, 2007; He et al., 2022). Components and concentration levels of organic compounds affect largely on atmospheric chemistry, atmospheric oxidation capacity, and radiation balance (Monks et al., 2015; Wu et al., 2020), as well as human health (Xing et al., 2018). According effective saturation concentrations (Donahue et al., 2012), organic compounds can be divided into intermediate volatile organic compounds (IVOCs), semi-volatile organic compounds (SVOCs), low volatile organic compounds (LVOCs), and extremely low volatile organic compounds (ELVOCs). Due to high yields of secondary organic aerosol (SOA) (Lim and Ziemann, 2009; Robinson et al., 2007), IVOCs have been proved to be important SOA precursors in urban atmospheres (Tkacik et al., 2012; Zhao et al., 2014).

Many studies showed that higher alkanes (i.e., linear and branched alkanes with 12-20 carbon atoms) to be important chemical components of IVOCs (Li et al., 2019; Zhao et al., 2014). Similar to these acyclic alkanes, cycloalkanes can also account for significant fractions of IVOCs. Cycloalkanes can reach more than 20% of IVOCs concentrations in diesel vehicle exhausts, lubricating oil, and diesel fuels (Alam et al., 2018; Liang et al., 2018; Lou et al., 2019), which are comparable or even higher than linear and branched alkanes. In some oil and gas regions, high concentrations of cycloalkanes were also reported (Aklilu et al., 2018; Gilman et al., 2013; Warneke et al., 2014). More importantly, laboratory studies suggest that SOA yields of cyclic and polycyclic alkanes are significantly higher than linear or branched alkanes with the same carbon number (as high as a factor of 5) (Hunter et al., 2014; Jahn et al., 2021; Li et al., 2021; Loza et al., 2014; Yee et al., 2013). As the result, cyclic and bicyclic species are shown to be large contributors to SOA formation potential from vehicles (Xu et al., 2020a, b; Zhao et al., 2015; Zhao et al., 2016). Recently, Hu et al. (2022) proposed that



IVOCs contributions to SOA formation in an urban region can increases from 8-20%
(acyclic alkanes only) to 17.5-46% if cycloalkanes are considered, signifying the
importance of cycloalkanes in SOA formation.
Cycloalkanes  are  mainly  measured  using  gas  chromatography-mass
spectrometer/flame ionization detector (GC-MS/FID) and two-dimensional gas
chromatography techniques (GC×GC) (Alam et al., 2018; Alam et al., 2016; de Gouw
et al., 2017; Liang et al., 2018; Zhao et al., 2016). Based on measurements of gas
chromatographic techniques, the signals of unspeciated cyclic compounds can be
determined from subtracting the total signal for each retention time bin according to the
series of $n$-alkanes (Zhao et al., 2014; Zhao et al., 2016). The mass of linear alkanes and
branched alkanes in each bin is calculated by using the total ion current (TIC) and the
fraction of characteristic fragments ($C_4H_9^+$, m/z 57) (Zhao et al., 2014; Zhao et al.,
2016). However, this type of quantitative method does not explicitly distinguish
individual cycloalkanes, and the determined mass may contain other cyclic compounds,
e.g., polycyclic aromatic hydrocarbons and compounds containing oxygen or
multifunctional groups (Zhao et al., 2014; Zhao et al., 2015; Zhao et al., 2016). Due to
the need for collection and pretreatment of air samples, time resolution of GC-MS
techniques is usually in the range of 0.5-1.0 h or above.
Proton transfer reaction mass spectrometer (PTR-MS) using hydronium ions
($H_3O^+$) as the reagent ion, is capable for measuring many organic compounds with high
response time and sensitivity (de Gouw and Warneke, 2007; Yuan et al., 2017).
However, detection of alkanes and cycloalkanes using PTR-MS with $H_3O^+$ ionization
is challenging, as usually only a series of fragment ions ($C_nH_{2n+1}^+$, $C_nH_{2n-1}^+$, $n \geqslant 3$) are
observed (Erickson et al., 2014; Gueneron et al., 2015). Recently, it was demonstrated
that linear alkanes can be measured by PTR-MS with time-of-flight detector using $NO^+$
as reagent ions ($NO^+$ PTR-ToF-MS) (Inomata et al., 2014; Koss et al., 2016; Wang et
al., 2020). These higher alkanes are ionized by $NO^+$ via hydride ion transfer leading to
major product ions of $C_nH_{2n+1}^+$, with low degree of fragmentation (Inomata et al., 2014).



Meanwhile, it is interesting that cycloalkanes were also tried to be quantified using
$C_nH_{2n+1}^+$ ions in $H_3O^+$ PTR-MS in oil and gas regions (Koss et al., 2017; Warneke et
al., 2014; Yuan et al., 2014), though sensitivities were substantially low (~10% of other
species) (Warneke et al., 2014). These evidences suggest that $NO^+$ ionization scheme
could provide a possibility for measuring cycloalkanes along with acyclic alkanes, as
demonstrated in a recent laboratory chamber work (Wang et al., 2022a).

In this study, we discuss the potential of online measurements of cycloalkanes in

ambient air and form emission sources utilizing $NO^+$ ionization in PTR-ToF-MS. The
results of laboratory experiments of characterization of product ions, calibration, and
response time will be shown. Finally, measurements of cycloalkanes using $NO^+$ PTR-
ToF-MS will be demonstrated from deployments at an urban site in southern China and
a chassis dynamometer study for vehicular emissions.
**2     Methods**
**2.1 $NO^+$ PTR-ToF-MS measurements**

A commercially PTR-ToF-MS instrument (Ionicon Analytik, Austria) equipped

with a quadrupole ion (Qi) guide for effective transfer of ions from drift tube to the
time-of-flight mass spectrometer is used for this work (Sulzer et al., 2014), and the mass
resolving approximately reach about 3000 m/$\Delta$m (Fig. S1). In order to generate $NO^+$
ions, 5 sccm ultra-high-purity air ($O_2+N_2 \geqslant 99.999\%$) is directed into to the hollow
cathode discharge area of ion source, $NO^+$ ions are produced by ionization as follows
(Federer et al., 1985):

$$O_2^+ + NO \rightarrow NO^+ + O_2 \qquad\qquad (a)$$

For the purpose of ionize $NO^+$ ions to the greatest extent, reduce the generation of

impurity ions such as $H_3O^+$, $O_2^+$ and $NO_2^+$, the ion source voltage Us and $U_{SO}$ were set
to 40 V and 100 V ,while drift tube voltages Udx and Udrift were set to 23.5 V and 470
V with drift tube pressure at 3.8 mbar, resulting in an *E/N* (electric potential intensity
relative to gas number density) of 60 Td (Wang et al., 2020). The measured ToF data is





processed for high-resolution peak fitting using Tofware (Tofwerk AG, version, 3.0.3),
obtaining high precision signals for cycloalkanes (Fig. S2). A description of the fitting
and calculation methods are fully discussed in previous studies (Stark et al., 2015;
Timonen et al., 2016). The raw ion count signals of $NO^+$ PTR-ToF-MS are normalized
to the primary ion ($NO^+$) at a level of $10^6$ cps to account for fluctuations of ion source
and detector (see SI).

Compared to proton transfer reactions between $H_3O^+$ ions and VOCs species, $NO^+$

ions show a variety of reaction pathways with VOCs, which can be roughly summarized
as follow:

charge transfer:

$$NO^+ + MH \rightarrow MH^+ + NO \qquad (b)$$

hydride ion transfer:

$$NO^+ + MH \rightarrow M^+ + HNO \qquad (c)$$

association reaction:

$$NO^+ + M + N_2 \rightarrow M \cdot (NO^+) + N_2 \qquad (d)$$

As shown in Fig. 1, the ionization energy (IE) of VOC species is a determination

factor for the reaction pathways with $NO^+$. For example, as the IE of NO is 9.26 eV
(Reiser et al., 1988), species with IE less than 9.26 eV, e.g., benzene and isoprene, will
undergo charge transfer reaction (b) with $NO^+$ (Španěl and Smith, 1999, 1996), while
species with IE greater than 9.26 eV, e.g., acetone and *n*-undecane, will undergo hydride
ion transfer (c) or association reaction (d) with $NO^+$ (Amador-Muñoz et al., 2016;
Diskin et al., 2002; Koss et al., 2016).
**2.2 Calibration and correction experiments**

In this study, we investigate characteristic ions of cycloalkanes produced undergo

the $NO^+$ ionization from a series of species identification experiments. The information
of cycloalkanes chemicals used in these experiments is listed in Table 1. In addition, we
also evaluated potential interferences from mono-alkenes, the isomers of cycloalkanes.
Calibration experiments were carried out to obtain sensitivities of cycloalkanes in both





the laboratory and the field, using a customized cylinder gas standard (Apel-Riemer
Environmental, Inc. USA), containing five different alkyl-cyclohexanes ($C_{10}$-$C_{14}$) and
eight *n*-alkanes ($C_8$-$C_{15}$) (Table S1).

Furthermore, some additional experiments were performed to explore the

influences of humidity and tubing delay effects on measurements of cycloalkanes.
Previously, it was shown that response factors of higher alkanes in $NO^+$ PTR-ToF-MS
are slightly affected by air humidity, and the degree of influence is related to carbon
number (Wang et al., 2020). Therefore, we evaluate the influence of humidity on
sensitivity of cycloalkanes in $NO^+$ PTR-ToF-MS using a custom-built humidity
delivery system (Fig. S3), and the results are applied to explore the relationship between
sensitivities of cycloalkanes and humidity. The perfluoroalkoxy (PFA) Teflon tubing is
used for inlets in this study, but gas-wall partitioning can be important for low volatility
compounds (Pagonis et al., 2017). As the result, measurements from controlled
laboratory experiments and field deployments were analyzed to systematically quantify
and characterize tubing delay time of cycloalkanes.
**3     Results and discussion**
**3.1 Characterization of product ion distribution**

$NO^+$ PTR-ToF-MS was used to directly measure high-purity cycloalkane

chemicals and identify the characteristic product ions produced by cycloalkanes under
$NO^+$ ionization. Here, the major product ions, fragments and their contributions for
different cycloalkanes are shown in Table 1. Chemical formulas of the product ions are
determined based on the positions of measured mass peaks.

Fig. 2 shows mass spectra within the relevant range (*m/z* $60^+$ to $200^+$ Th) for

cycloalkanes. The signals are normalized to the largest ion peak for better comparison.
As shown in Fig. 2, no significant fragmentation appears for cycloheptane and
methylcyclohexane ($C_7H_{14}$), and the dominating product ions are observed at *m/z* 97
Th,   corresponding   to   $C_7H_{13}^+$.   Similarly,   the   product   ions   generated   by





hexylcyclohexane and cyclododecane ($C_{12}H_{24}$) under $NO^+$ ionization mainly appear at
$m/z$ 167 Th (Fig. 2c-d), corresponding to $C_{12}H_{23}^+$, and fragments occurred at $m/z$ 97 Th
and $m/z$ 111 Th, corresponding to $C_7H_{13}^+$ and $C_8H_{15}^+$, respectively. Bicyclic
cycloalkanes undergo the same ionization channel from $NO^+$ ionization, as
demonstrated by major product ions at $m/z$ 165 Th ($C_{12}H_{21}^+$) and other fragmentation
ions from bicyclohexyl ($C_{12}H_{22}$) (Fig. 2e). These results verify that reactions of cyclic
and bicyclic alkanes with $NO^+$ ions follow the hydride ion transfer pathway to yield
$C_nH_{2n-1}^+$ and $C_nH_{2n-3}^+$ product ions, respectively. As mentioned above, the characteristic
peaks of cycloalkanes under $NO^+$ ionization is consistent with the ions for attempts to
utilize $H_3O^+$ PTR-MS for cycloalkanes detection in previous studies, though with low
sensitivities reported for cycloalkanes (Erickson et al., 2014; Gueneron et al., 2015;
Warneke et al., 2014; Yuan et al., 2014). As the result, we speculate that reactions of
$NO^+$ with cycloalkanes may present large contributions to cycloalkane M-H product
ions in the $H_3O^+$ chemistry mode of PTR-MS (Španěl et al., 1995).

As the isomers of cyclic alkanes, alkenes may interfere with measurements of

cycloalkanes. Here, we use 1-heptene ($C_7H_{14}$) and 1-decene ($C_{10}H_{20}$), isomers of $C_7$ and
$C_{10}$ cyclic alkanes, to explore the ionization regime of alkenes in $NO^+$ chemical
ionization (Fig. 3 and Table 1). These two alkenes produce more fragments than
cycloalkanes under $NO^+$ chemistry, but mainly react with $NO^+$ via association reaction
to yield $C_nH_{2n}\bullet(NO)^+$ product ions. The major product ions of 1-heptene and 1-decene
appear at $m/z$ 128 Th and $m/z$ 170 Th, corresponding to $C_7H_{13}HNO^+$ and $C_{10}H_{19}HNO^+$,
respectively. Based on the mass spectra, alkenes produce the $C_nH_{2n-1}^+$ product ions at
fractions of <5%, which are similar to $NO^+$ ionization results of the two species and
other 1-alkenes determined from a selected ion flow tube mass spectrometer (SIFT-MS)
(Diskin et al., 2002). The same study (Diskin et al., 2002) also demonstrated that trans-
2-alkenes might produce more $C_nH_{2n-1}^+$ ions under $NO^+$ ionization (e.g., trans-2-
heptene contributions 40% of $C_7H_{13}^+$ ions). However, concentrations of 1-alkenes and
trans-2-alkenes in the atmosphere are usually significantly lower than cycloalkanes





(about 25% and <5%, respectively) (de Gouw et al., 2017; Yuan et al., 2013). Therefore,
we conclude the interferences of alkenes on measurements of cycloalkanes in most
environments are small.

## 3.2 Sensitivity, humidity dependence and detection limits

The calibration experiments of cycloalkanes are carried out in both dry conditions

(<1% RH) and humidified conditions (Fig. S5). Fig. 4 illustrates results from a typical
calibration experiment for five different alkyl-substituted cyclohexanes with carbon
atoms of 10-14 in dry air (relative humidity < 1%) for $NO^+$ PTR-ToF-MS. There is a
good linear relationship between ion signals and concentrations of various cycloalkanes
(R=0.9996-0.9999). Sensitivities (ncps $ppb^{-1}$) of cycloalkanes are in the range of 210 to
260 ncps $ppb^{-1}$ (Table 2). The sensitivity of each cycloalkanes remained stable in the
long-term calibration conducted in the laboratory and in the field (Fig. S6). Table 2
further shows detection limits of cycloalkanes by $NO^+$ PTR-ToF-MS, which are
calculated as the concentrations with the signal-to-noise ratio of 3 (Bertram et al., 2011;
Yuan et al., 2017). The detection limits of cycloalkanes at integral time of 10 s and 1
min are in the range of 6.2 to 8.2 ppt and 2.4-3.6 ppt, respectively. For comparison, the
detection limits of $NO^+$ PTR-ToF-MS for 24 h of integral time are also determined,
obtaining comparable results (<0.1 ppt) with detection limits of GC×GC-ToF-MS (0.1-
0.3 ppt for daily sample) (Liang et al., 2018; Xu et al., 2020a),.

Fig. 4b shows that normalized signals of hexylcyclohexane relative to dry

conditions as a function of different humidity. The relative signals of the explored
cycloalkanes show minor decrease (<10%) at the highest humidity (~82% RH at 25℃)
compare to dry condition, and the observed changes for cycloalkanes with different
carbon number are similar, suggesting little influence of humidity on measurements of
cycloalkanes. The humidity-dependence curves determined in Fig. 4b are used to
corrected variations of ambient humidity in the atmosphere.

The response time of various cycloalkanes in the instrument and also sampling

tubing is determined from laboratory experiments (Fig. S6). For the species not in the



gas standard, we also take advantage of vehicular emissions measurements associated
with high concentrations of cycloalkanes. Here, we use the delay time to determine
response of cycloalkanes, which is calculated based on the time it takes for signals to
drop to 10% of its initial value (Fig. S7) (Pagonis et al., 2017). The delay time of
cycloalkanes are summarized in Fig. 5. The delay time of various cycloalkanes
generally increases with the carbon numbers, ranging from a few seconds to a few
minutes, but relatively lower than those acyclic alkanes within $C_{10}$-$C_{15}$ range (Wang et
al., 2020). These results suggest that measured variability of cycloalkanes with higher
carbon number, especially for $C_{19}$-$C_{20}$ or above, may only be reliable for time scales
longer than 10 min.
**3.3 Applications in ambient air and vehicular exhausts**
Based on the results shown above, we deployed the $NO^+$ PTR-ToF-MS to measure
concentration levels and variations of cycloalkanes at an urban site in Guangzhou,
southern China. Details of the field campaign were described in previous studies (Wang
et al., 2020; Wu et al., 2020). The average sensitivities of long-term calibration (dry
condition) during the field observations were used to quantify cycloalkanes after
corrected variations of ambient humidity in the atmosphere. For the cycloalkanes that
are not contained in the gas standard, we employ average sensitivity for calibrated
cycloalkanes in gas standard.
The concentration levels and diurnal profiles of cyclic and bicyclic alkanes are
illustrated in Fig. 6a, along with CO. In general, cyclic and bicyclic alkanes
demonstrated similar temporal variability as CO, suggesting cyclic and bicyclic alkanes
may be mainly emitted from primary combustion sources. Concentrations of $C_{12}$
bicyclic alkanes are observed to be comparable with $C_{12}$ cyclic alkanes. As shown in
Fig. 6b, selected $C_{10}$ and $C_{12}$ cycloalkanes show diurnal variations with lower
concentration during the daytime. Compared to diurnal variations of other species with
different reactivity (Wu et al., 2020), the decline fractions of cycloalkanes are more
comparable to reactive species (e.g., $C_8$ aromatics) than the inert ones (e.g., CO,



benzene), indicating significantly daytime photochemical removal of these
cycloalkanes. Diurnal patterns of other cyclic and bicyclic alkanes exhibit similar
results (Fig. S8). As discussed in Wang et al. (2020), similar diurnal profiles of
cycloalkanes with different carbon number also imply that tubing-delay effect may not
affect significantly to temporal variations of cycloalkanes reported here. Based on both
time series and correlation analysis (Fig. 6c), cyclic and bicyclic alkanes showed strong
correlation with acyclic alkanes, suggesting they came from same emission sources.

NO$^+$ PTR-ToF-MS was also applied to measure cycloalkanes emissions along with

other organic compounds from vehicles using gasoline, diesel, and LPG as fuel, by
conducting a chassis dynamometer measurement (Wang et al., 2022b). Fig. 7 shows
time series of C$_{12}$ cyclic and bicyclic alkanes, C$_{12}$ alkanes, toluene, and acetaldehyde
measured by NO$^+$ PTR-ToF-MS for a gasoline vehicle with emission standard of China
III and a diesel vehicle with emission standard of China IV. Both test vehicles were
started with hot engines. As shown in Fig. 7a, high concentrations of selected
cycloalkanes emitted by the gasoline vehicle were observed as the engine started.
Compared with typical VOC compounds exhausted by vehicles (e.g., toluene and
acetaldehyde), concentrations of cycloalkanes were lower but showed similar temporal
variations. In comparison, cycloalkanes emissions from diesel vehicles are obviously
different. As shown in Fig. 7b, concentrations of cycloalkanes showed relatively
moderate enhancements as engine started, but significantly enhanced with high vehicle
speed, obtaining periodic pattern variations within each test cycle. Though the highest
concentrations of cycloalkanes observed for gasoline and diesel vehicles are similar,
determined emission factors of diesel vehicles are significantly larger than gasoline
vehicles, since emissions of cycloalkanes are mainly concentrated during a short period
at engine start for gasoline vehicles, whereas emissions of cycloalkanes remain high
during hot running for diesel vehicles. For instance, the determined emission factors of
C$_{12}$ cyclic alkanes are 0.06 mg km$^{-1}$ for gasoline vehicle and 1.17 mg km$^{-1}$ for diesel
vehicles, respectively. Recent studies reported that cyclic compounds occupied a large



proportion in IVOCs emitted from vehicles, with prominent SOA formation potentials
(Fang et al., 2021; Huang et al., 2018; Zhao et al., 2016), but emissions of individual
cycloalkanes were not reported yet. As the result, high time-resolution measurements
of cycloalkanes from vehicular emissions by $NO^+$ PTR-ToF-MS can improve the
characterization of emission mechanisms of these species.

**3.4 Insights from simultaneous measurements of cycloalkanes and**

**296 alkanes**


Since $NO^+$ PTR-ToF-MS can provide simultaneous measurements of cycloalkanes
and acyclic alkanes, we can use this information to explore the relative importance of
cycloalkanes. Fig. 8 shows mean concentrations of cycloalkanes (cyclic and bicyclic)
and alkanes with different carbon numbers ($C_{10}$-$C_{20}$) measured by $NO^+$ PTR-ToF-MS
in urban air and emissions from diesel vehicles. In urban region, concentrations of
cyclic and bicyclic cycloalkanes are comparable, but lower than acyclic alkanes, with
concentration ratios relative to acyclic alkanes in the range of 0.30-0.46 for cyclic
alkanes and 0.23-0.51 for bicyclic alkanes. Similar results are obtained for gasoline
vehicles, with cyclic alkanes/acyclic alkanes and bicyclic alkanes/acyclic ratios of 0.27-
0.53 for and 0.21-0.52, respectively (Fig. 9). The contributions of cycloalkanes in diesel
vehicular emissions are relatively higher, with the concentration ratios relative to
acyclic alkanes in the range of 0.42-0.66 for cyclic alkanes and 0.37-0.95 for bicyclic
alkanes.
As there is only limited measurements of bicyclic alkanes in the literature, we
compare concentration ratios of cyclic alkanes to acyclic alkanes with results in
previous studies, mainly determined from measurements by GC-MS/FID and GC×GC.
As shown in the Fig. 9, the ratios obtained in the urban region of Guangzhou in this
work (0.2-0.4) are similar to other measurements in urban area, including London, UK
(Xu et al., 2020b) and Algiers, Algeria (Yassaa et al., 2001). As for emissions from
diesel vehicles, the ratios measured in this study are similar to GC×GC measurements
in UK (Alam et al., 2016) for $C_{12}$-$C_{14}$ range, whereas the ratios from this study are





higher than Alam et al. (2016) for larger carbon number. The relative fractions of
cycloalkanes measured from oil and gas region (Gilman et al., 2013) and emissions
from lubricating oil (Liang et al., 2018) (>0.7) are relatively higher than ambient air
and vehicular emissions. The variability pattern of cyclic alkanes to acyclic alkanes
ratios for different carbon number may be used for source analysis of these IVOCs in
the future.
**4     Conclusion**

In this study, we show that online measurements of cycloalkanes can be achieved

using proton transfer reaction time-of-flight mass spectrometry with $NO^+$ chemical
ionization ($NO^+$ PTR-ToF-MS). Our results demonstrate that cyclic and bicyclic
alkanes are ionized via hydride ion transfer leading to characteristic product ions of
$C_nH_{2n-1}^+$ and $C_nH_{2n-3}^+$, respectively. As isomers of cycloalkanes, alkenes undergoes
association reactions mainly yielding product ions $C_nH_{2n} \cdot (NO)^+$, which induce little
interference to cycloalkanes detection. Calibration of various cycloalkanes show
similar sensitivities with small humidity dependence. The detection limits of
cycloalkanes are in the range of 2-4 ppt at integral time of 1 min.

Appling this method, cycloalkanes were successfully measured at an urban site

in southern China and a chassis dynamometer study for vehicular emissions.
Concentrations of both cyclic and bicyclic alkanes are substantial in urban air and
vehicular emissions. Diurnal variations of cycloalkanes in the urban air indicate
significant losses due to photochemical processes during the daytime. The
concentration ratios of cyclic alkanes to acyclic alkanes are similar between urban air
and gasoline vehicle emissions, but higher for diesel vehicles, which could be
potentially used for source analysis in future studies. Our work demonstrates that $NO^+$
PTR-ToF-MS provides a new complementary approach for fast characterization of
cycloalkanes in both ambient air and emission sources, which can be helpful to
investigate sources of cycloalkanes and their contribution to SOA formation in the



atmosphere. Measurements of cycloalkanes in the particle phase may also be possible
by combining NO$^+$ PTR-ToF-MS with "chemical analysis of aerosols online"
(CHARON) or other similar aerosol inlets (Muller et al., 2017), which could further
provide particle-phase information of cycloalkanes and gas-partitioning analysis of
cycloalkanes.

**Data availability**

Data are available from the authors upon request.

**Author contribution**

BY designed the research. YBC, CMW, SHW, XJH, CHW, XS, YBH, XBL, YJL

and MS contributed to data collection. YBC performed data analysis, with contributions
from CMW, SHW, XJH, and CHW. YBC and BY prepared the manuscript with
contributions from other authors. All the authors reviewed the manuscript.
**Competing interests**

The authors declare that they have no known competing financial interests or

personal relationships that could have appeared to influence the work reported in this
paper.
**Acknowledgement**

This work was supported by the National Natural Science Foundation of China

(grant No. 41877302, 42121004), Key-Area Research and Development Program of
Guangdong Province (grant No. 2020B1111360003), and Guangdong Innovative and
Entrepreneurial Research Team Program (grant No. 2016ZT06N263). This work was
also supported by Special Fund Project for Science and Technology Innovation Strategy
of Guangdong Province (Grant No.2019B121205004).

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

Methods to extract molecular and bulk chemical information from series of complex
mass spectra with limited mass resolution, International Journal of Mass Spectrometry,





389, 26-38, 2015.

Sulzer, P., Hartungen, E., Hanel, G., Feil, S., Winkler, K., Mutschlechner, P.,

Haidacher, S., Schottkowsky, R., Gunsch, D., Seehauser, H., Striednig, M., Jürschik, S.,
Breiev, K., Lanza, M., Herbig, J., Märk, L., Märk, T. D., and Jordan, A.: A Proton
Transfer Reaction-Quadrupole interface Time-Of-Flight Mass Spectrometer (PTR-
QiTOF): High speed due to extreme sensitivity, International Journal of Mass
Spectrometry, 368, 1-5, 2014.

Timonen, H., Cubison, M., Aurela, M., Brus, D., Lihavainen, H., Hillamo, R.,

Canagaratna, M., Nekat, B., Weller, R., Worsnop, D., and Saarikoski, S.: Applications
and limitations of constrained high-resolution peak fitting on low resolving power mass
spectra from the ToF-ACSM, Atmospheric Measurement Techniques, 9, 3263-3281,

2016.

Tkacik, D. S., Presto, A. A., Donahue, N. M., and Robinson, A. L.: Secondary

Organic Aerosol Formation from Intermediate-Volatility Organic Compounds: Cyclic,
Linear, and Branched Alkanes, Environmental Science & Technology, 46, 8773-8781,

2012.

Wang, C. M., Yuan, B., Wu, C. H., Wang, S. H., Qi, J. P., Wang, B. L., Wang, Z.

L., Hu, W. W., Chen, W., Ye, C. S., Wang, W. J., Sun, Y. L., Wang, C., Huang, S., Song,
W., Wang, X. M., Yang, S. X., Zhang, S. Y., Xu, W. Y., Ma, N., Zhang, Z. Y., Jiang, B.,
Su, H., Cheng, Y. F., Wang, X. M., and Shao, M.: Measurements of higher alkanes using
NO$^+$ chemical ionization in PTR-ToF-MS: important contributions of higher alkanes to
secondary organic aerosols in China, Atmospheric Chemistry and Physics, 20, 14123-

14138, 2020.

Wang, K., Wang, W., Fan, C., Li, J., Lei, T., Zhang, W., Shi, B., Chen, Y., Liu, M.,

Lian, C., Wang, Z., and Ge, M.: Reactions of C12-C14 n-Alkylcyclohexanes with Cl
Atoms: Kinetics and Secondary Organic Aerosol Formation, Environmental Science &
Technology, 56, 4859-4870, 2022a.

Wang, S., Yuan, B., Wu, C., Wang, C., Li, T., He, X., Huangfu, Y., Qi, J., Li, X. B.,



Sha, Q., Zhu, M., Lou, S., Wang, H., Karl, T., Graus, M., Yuan, Z., and Shao, M.:
Oxygenated volatile organic compounds (VOCs) as significant but varied contributors
to VOC emissions from vehicles, Atmospheric Chemistry and Physics, 22, 9703-9720,
2022b.
Warneke, C., Geiger, F., Edwards, P. M., Dube, W., Pétron, G., Kofler, J., Zahn, A.,
Brown, S. S., Graus, M., Gilman, J. B., Lerner, B. M., Peischl, J., Ryerson, T. B., de
Gouw, J. A., and Roberts, J. M.: Volatile organic compound emissions from the oil and
natural gas industry in the Uintah Basin, Utah: oil and gas well pad emissions compared
to ambient air composition, Atmospheric Chemistry and Physics, 14, 10977-10988,

2014.

Wu, C., Wang, C., Wang, S., Wang, W., Yuan, B., Qi, J., Wang, B., Wang, H., Wang,
C., Song, W., Wang, X., Hu, W., Lou, S., Ye, C., Peng, Y., Wang, Z., Huangfu, Y., Xie,
Y., Zhu, M., Zheng, J., Wang, X., Jiang, B., Zhang, Z., and Shao, M.: Measurement
report: Important contributions of oxygenated compounds to emissions and chemistry
of volatile organic compounds in urban air, Atmospheric Chemistry and Physics, 20,

14769-14785, 2020.

Xing, L. Q., Wang, L. C., and Zhang, R.: Characteristics and health risk assessment
of volatile organic compounds emitted from interior materials in vehicles: a case study
from Nanjing, China, Environmental Science and Pollution Research, 25, 14789-14798,

2018.

Xu, R., Alam, M. S., Stark, C., and Harrison, R. M.: Behaviour of traffic emitted
semi-volatile and intermediate volatility organic compounds within the urban
atmosphere, Science of The Total Environment, 720, 2020a.
Xu, R., Alam, M. S., Stark, C., and Harrison, R. M.: Composition and emission
factors of traffic- emitted intermediate volatility and semi-volatile hydrocarbons ($C_{10}$–
$C_{36}$) at a street canyon and urban background sites in central London, UK, Atmospheric
Environment, 231, 2020b.
Yassaa, N., Meklati, B. Y., Brancaleoni, E., Frattoni, M., and Ciccioli, P.: Polar and



non-polar volatile organic compounds (VOCs) in urban Algiers and saharian sites of
Algeria, Atmospheric Environment, 35, 787-801, 2001.
Yee, L. D., Craven, J. S., Loza, C. L., Schilling, K. A., Ng, N. L., Canagaratna, M.
R., Ziemann, P. J., Flagan, R. C., and Seinfeld, J. H.: Effect of chemical structure on
secondary organic aerosol formation from $C_{12}$ alkanes, Atmospheric Chemistry and
Physics, 13, 11121-11140, 2013.
Yuan, B., Hu, W. W., Shao, M., Wang, M., Chen, W. T., Lu, S. H., Zeng, L. M.,
and Hu, M.: VOC emissions, evolutions and contributions to SOA formation at a
receptor site in eastern China, Atmospheric Chemistry and Physics, 13, 8815-8832,

2013.

Yuan, B., Koss, A. R., Warneke, C., Coggon, M., Sekimoto, K., and de Gouw, J.
A.: Proton-Transfer-Reaction Mass Spectrometry: Applications in Atmospheric
Sciences, Chem Rev, 117, 13187-13229, 2017.
Yuan, B., Warneke, C., Shao, M., and de Gouw, J. A.: Interpretation of volatile
organic compound measurements by proton-transfer-reaction mass spectrometry over
the deepwater horizon oil spill, International Journal of Mass Spectrometry, 358, 43-48,

2014.

Zhao, Y., Hennigan, C. J., May, A. A., Tkacik, D. S., De Gouw, J. A., Gilman, J.
B., Kuster, W. C., Borbon, A., and Robinson, A. L.: Intermediate-Volatility Organic
Compounds: A Large Source of Secondary Organic Aerosol, Environmental Science &
Technology, 48, 13743-13750, 2014.
Zhao, Y., Nguyen, N. T., Presto, A. A., Hennigan, C. J., May, A. A., and Robinson,
A. L.: Intermediate Volatility Organic Compound Emissions from On-Road Diesel
Vehicles: Chemical Composition, Emission Factors, and Estimated Secondary Organic
Aerosol Production, Environmental Science & Technology, 49, 11516-11526, 2015.
Zhao, Y., Nguyen, N. T., Presto, A. A., Hennigan, C. J., May, A. A., and Robinson,
A. L.: Intermediate Volatility Organic Compound Emissions from On-Road Gasoline
Vehicles and Small Off-Road Gasoline Engines, Environmental Science & Technology,



597     50, 4554-4563, 2016.





**Table 1**. The formula, purity, ionization energy (IE) of the chemicals used in product
ion characterization experiments are shown. The percentage of each product ion from
the reactions with NO⁺ ions is indicated in brackets, and the major product ions are
identified in bold.

| Chemicals | Formula | Purity (%) | IE [a] (eV) | Product ions (%) | | |
|---|---|---|---|---|---|---|
| Cycloheptane | $C_7H_{14}$ | 98.0% | 9.82 | **$C_7H_{13}^+$(100)** | | |
| Methylcyclohexane | $C_7H_{14}$ | 99.0% | 9.64 | **$C_7H_{13}^+$(100)** | | |
| Cyclododecane | $C_{12}H_{24}$ | 99.0% | 9.72 | **$C_{12}H_{23}^+$(82)** | $C_8H_{15}^+$(8) | $C_7H_{13}^+$(10) |
| Hexylcyclohexane | $C_{12}H_{24}$ | 98.0% | N/A[b] | **$C_{12}H_{23}^+$(79)** | $C_8H_{15}^+$(10) | $C_7H_{13}^+$(11) |
| Bicyclohexyl | $C_{12}H_{22}$ | 99.0% | 9.41 | **$C_{12}H_{21}^+$(71)** | $C_5H_{11}^+$(17) | $C_5H_{13}^+$(<5) |
| | | | | $C_7H_{13}^+$(5) | $C_8H_{15}^+$(<5) | $C_8H_{15}^+$(<5) |
| 1-Heptene | $C_7H_{14}$ | 99.5% | 9.34 | **$C_7H_{13}HNO^+$(40)** | $C_5H_9HNO^+$(15) | $C_{12}H_{22}^+$(<5) |
| | | | | $C_3H_5HNO^+$(<5) | $C_7H14^+$(<5) | $C_4H_7HNO^+$(37) |
| 1-Decene | $C_{10}H_{20}$ | 99.5% | 9.42 | **$C_{10}H_{19}HNO^+$(51)** | $C_5H_9HNO^+$(18) | $C_6H_{11}HNO^+$(15) |
| | | | | $C_4H_7HNO^+$(12) | $C_{10}H_{19}^+$(<5) | $C_7H_{13}HNO^+$(<5) |
| | | | | $C_{10}H_{20}^+$(<5) | | |

[a] NIST chemistry web book (http://webbook.nist.gov)
[b] N/A stands for "not available"





**Table 2.** Carbon numbers and formula, means normalized sensitivities and detection
limits of cycloalkanes in $NO^+$ PTR-ToF-MS.

| Cycloalkanes (C number) | Formula | Normalized sensitivities (ncps ppb$^{-1}$) | Detection limit (ppt) | |
|---|---|---|---|---|
| | | | 10s | 1min |
| $C_{10}$ | $C_{10}H_{20}$ | 231.3 | 7.20 | 3.04 |
| $C_{11}$ | $C_{11}H_{22}$ | 207.8 | 7.72 | 2.76 |
| $C_{12}$ | $C_{12}H_{24}$ | 223.9 | 7.01 | 2.85 |
| $C_{13}$ | $C_{13}H_{26}$ | 244.6 | 6.24 | 2.46 |
| $C_{14}$ | $C_{14}H_{28}$ | 247.9 | 6.22 | 2.40 |
| $C_{15}$ | $C_{15}H_{30}$ | N/A[a] | 6.67 | 2.54 |
| $C_{16}$ | $C_{16}H_{32}$ | N/A | 7.28 | 2.96 |
| $C_{17}$ | $C_{17}H_{34}$ | N/A | 7.46 | 3.05 |
| $C_{18}$ | $C_{18}H_{36}$ | N/A | 7.90 | 3.40 |
| $C_{19}$ | $C_{19}H_{38}$ | N/A | 8.21 | 3.61 |
| $C_{20}$ | $C_{20}H_{40}$ | N/A | 8.08 | 3.48 |

[a] N/A stands for "not available ".



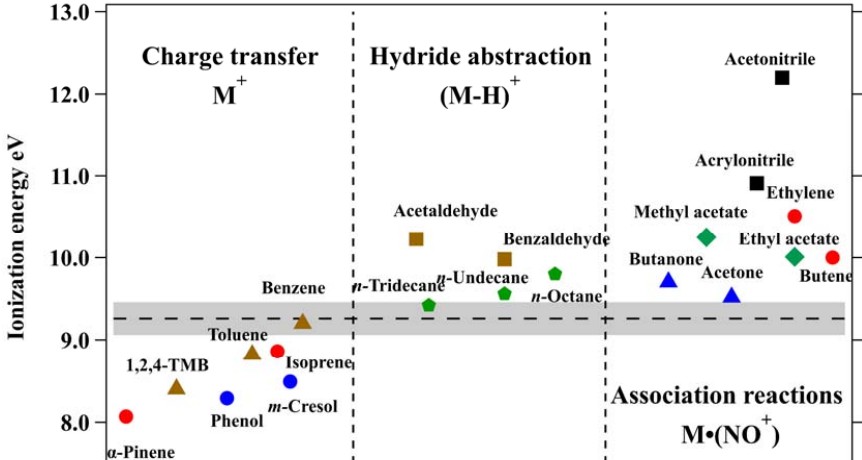

**Figure 1.** Ionization energy and reaction pathways with $NO^+$ ions of organic compounds including alkanes (green pentagon), aromatics (brown triangle), alkenes (red circle), phenolic species (blue circle), aldehydes (brown square), ketones (blue triangle), esters (green diamond), and nitrogen-containing species (black square). The ionization energy of NO (9.26 eV) is represented by the dashed line with shading representing reported uncertainty. The IE of various organic compounds are obtained in the NIST chemistry web book (http://webbook.nist.gov).



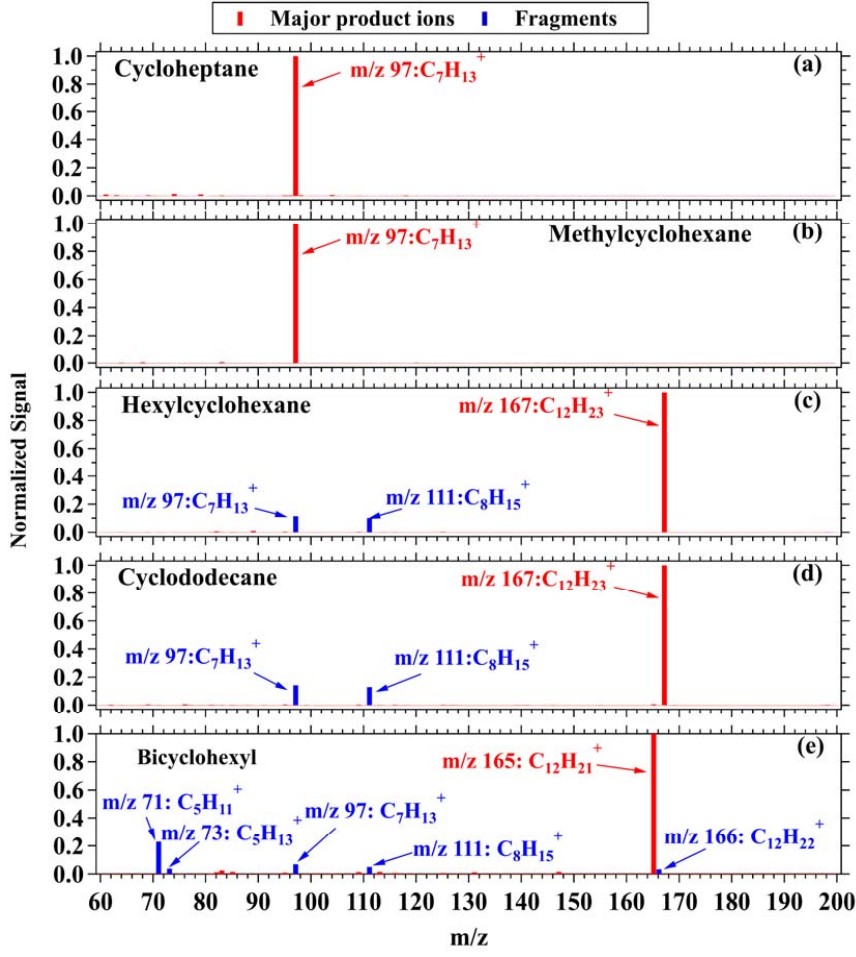

**Figure 2.** Mass spectra of product ions from cycloheptane **(a)**, methylcyclohexane **(b)**, hexylcyclohexane **(c)**, cyclododecane **(d)** and bicyclohexyl **(e)** in NO$^+$ PTR-ToF-MS. The major product ions are shown in red, and the fragments are shown in blue.





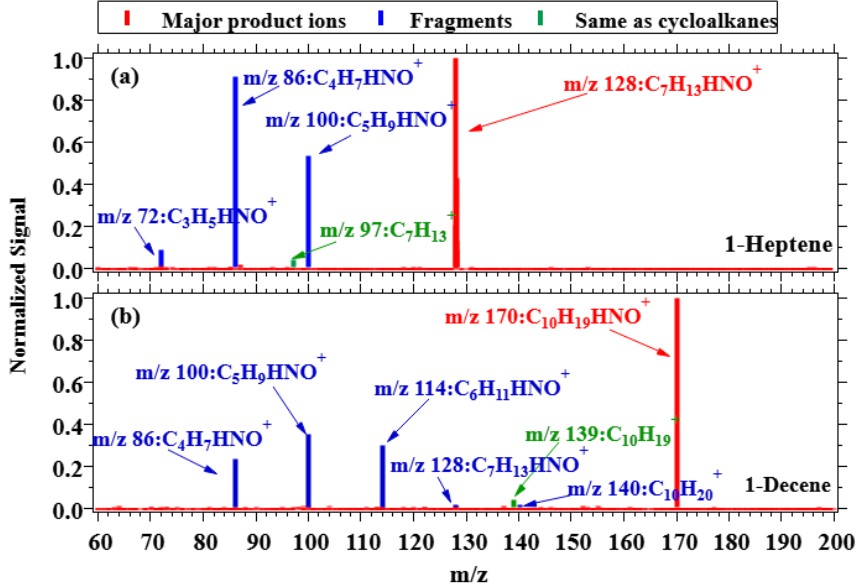

620

**Figure 3.** Mass spectra of product ions from 1-heptene **(a)**, and 1-decene **(b)** with NO$^+$

PTR-ToF-MS. The major product ions are shown in red. The same product ions as the

cycloalkanes (M-H ions) are shown in blue, and other fragments are shown in green.

624

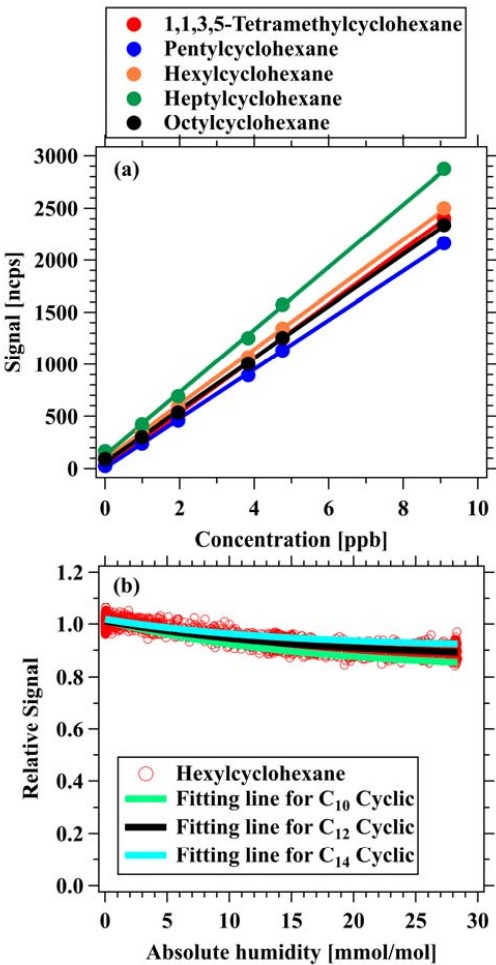

**Figure 4. (a)** Multipoint calibration curve for 1,1,3,5-tetramethylcyclohexane (red), pentylcyclohexane (bule), hexylcyclohexane (orange), heptylcyclohexane (green) and octylcyclohexane (black). **(b)** Humidity dependence of sensitivity for various cycloalkanes, including measurement results for hexylcyclohexane (red markers), and the fitted lines for $C_{10}$ cyclic alkane (green), $C_{12}$ cyclic alkane (black), and $C_{14}$ cyclic alkane (blue), with the corresponding fitted functions of $y=0.82+0.19\times\exp(-0.06x)$, $y=0.87+0.14\times\exp(-0.06x)$, and $y=0.90+0.11\times\exp(-0.07x)$, respectively.





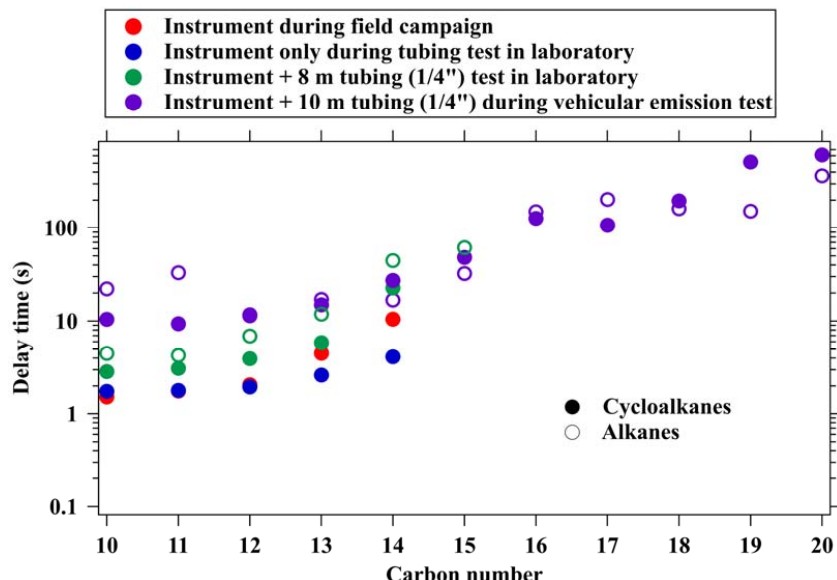

**Figure 5.** Delay time of cycloalkanes determined from measurements in the field, from laboratory experiments, and vehicular emissions. The delay times of alkanes from Wang et al. (2020) are also shown for comparison.

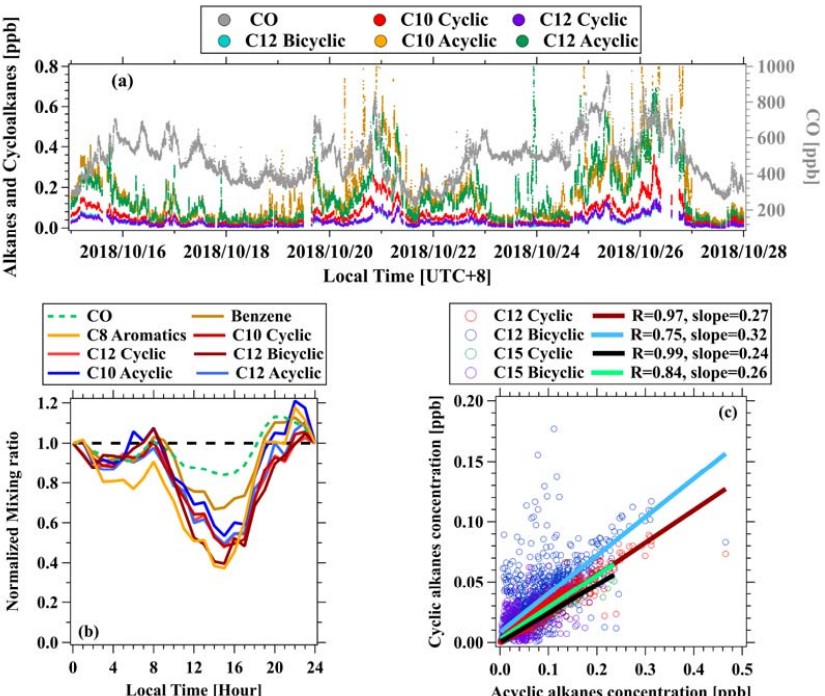

**Figure 6. (a)** Time series of CO, cyclic, bicyclic, and acyclic alkanes measured at the urban site in Guangzhou. **(b)** Normalized diurnal variations of CO, benzene, $C_8$ aromatics, $C_{10}$ cyclic alkanes, $C_{10}$ acyclic alkanes, $C_{12}$ cyclic alkanes, $C_{12}$ bicyclic alkanes and $C_{12}$ acyclic alkanes. The measurement data for each species is normalized to midnight concentrations. **(c)** Scatterplots of cyclic and bicyclic alkanes to acyclic alkanes with carbon atoms of 12 and 15.

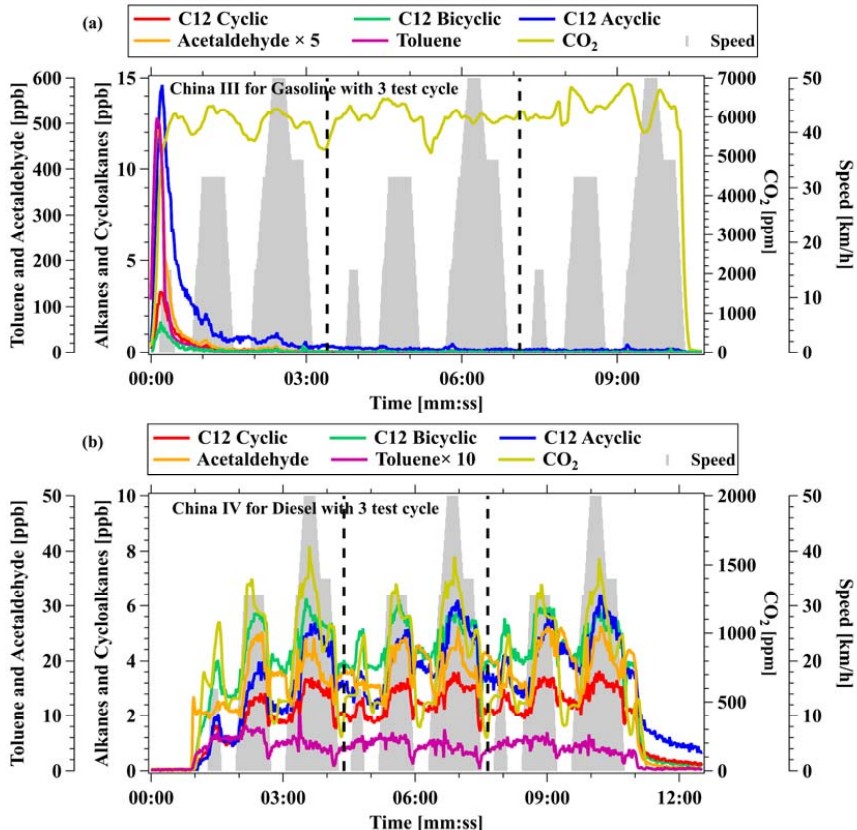


**Figure 7.** Concentrations of $C_{12}$ cyclic, bicyclic, and acyclic alkanes, acetaldehyde,
toluene, and $CO_2$ for **(a)** a gasoline vehicle with emission standard of China III and **(b)**
a diesel vehicle with emission standard of China IV. The gray shadows represent the
speeds of the vehicles on the chassis dynamometer.

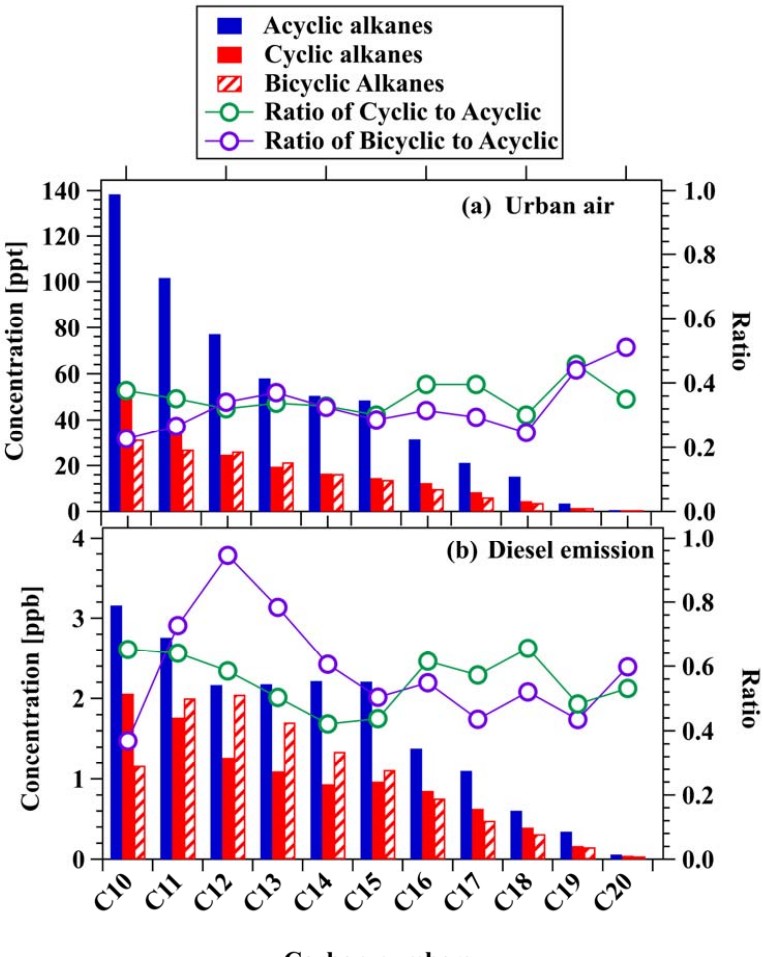

**Figure 8.** Mean concentrations of cyclic and bicyclic alkanes and alkanes (branched +

linear) with different carbon numbers measured by NO[+] PTR-ToF-MS in the urban air

**(a)** and diesel emissions **(b)**. The green and purple lines with circles represent the ratios

of cyclic and bicyclic alkanes to acyclic alkanes under the same carbon numbers,

respectively.

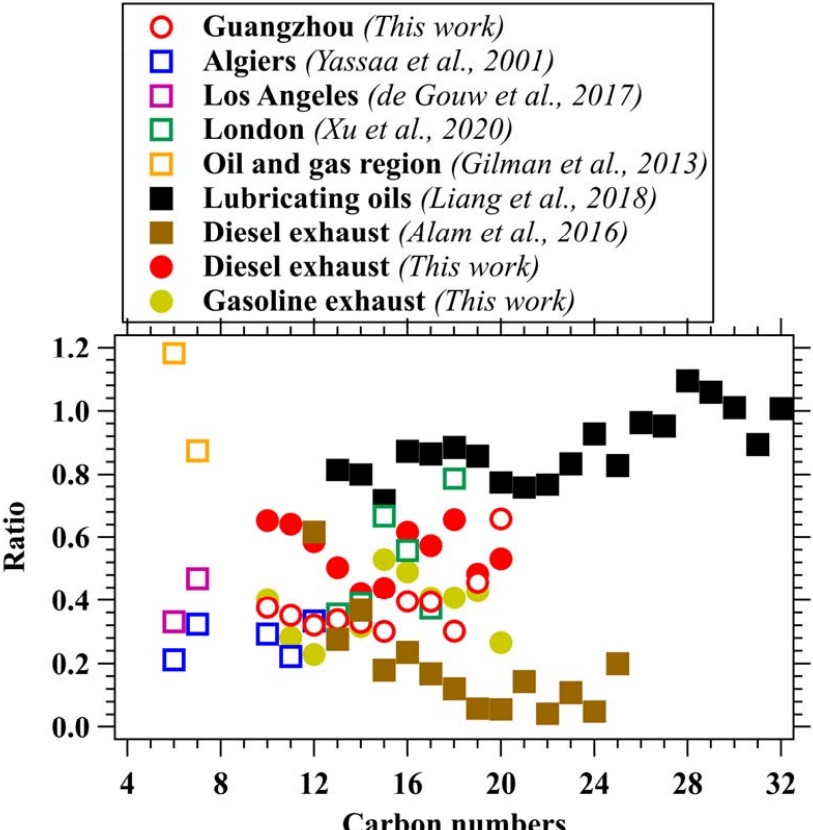

**Figure 9.** The concentrations ratios of cyclic alkanes to acyclic alkanes for different
carbon number. Measurements in various urban areas, including Guangzhou in China,
London in UK (Xu et al., 2020b), Los Angeles in US (de Gouw et al., 2017), Algiers in
Algeria (Yassaa et al., 2001), and an oil and gas region in Colorado of US (Gilman et
al., 2013) are also shown for comparison. Emission sources, including vehicle exhausts
(Alam et al., 2016) and lubricating oils (Liang et al., 2018) are also included.