# Peer review of "Online measurements of cycloalkanes based on NO[+] chemical ionization in proton transfer reaction time of flight mass spectrometry (PTR-ToF-MS)"

_EGUsphere, 2022_

## Referee Comment (RC1)

Review of "Online measurements of cycloalkanes based on NO$^+$ chemical ionization in proton transfer reaction time of flight mass spectrometry (PTR-ToF-MS)" by Chen et al. (egusphere-2022-880)

**General comments:**

This paper describes a new technique of online measurements of cycloalkanes in the atmosphere by means of NO$^+$ chemical ionization mass spectrometry. The authors used major product ions of $C_nH_{2n-1}^+$ and $C_nH_{2n-3}^+$ from NO$^+$ chemical ionization of cyclic and bicyclic alkanes, respectively, for the measurement. After characterizing this technique in the laboratory, this technique was applied for field measurements at an urban site in southern China and a chassis dynamometer study regarding vehicle emissions. I think that this paper is generally well-written. However, I can't avoid a sense of apprehension because the authors showed the results of C10-C20 cyclic and bicyclic alkanes based on mass spectra of only five C7 and C12 cycloalkanes. The authors should show, for example, each mass spectrum of C10-C20 cycloalkanes (e. g., n-alkylcyclohexane) and discuss about interferences of other species on the ion signals of the cycloalkanes before applying the technique to the filed measurements and the chassis dynamometer study. I think that the characterizations shown in this paper are not enough. In addition, I am not sure that there is no interference on the data obtained in the filed measurements and in the chassis dynamometer study. Therefore, I recommend this paper to be revised referring to my specific comments listed below.

**Specific comments:**

(1) Page 5, Line 123: I would like to know ion intensities of $O_2^+$ and $NO_2^+$ relative to that of NO$^+$ in a flow of zero (VOC-free) -air. I guess that the $O_2^+$ was consumed by reactions of VOCs in a sample air. The authors should evaluate the interference of the $O_2^+$ reactions on the signals of cycloalkanes.

(2) Page 6, Line 125: I would like to know ion signals at 166.18 Th ($C_{12}H_{22}^+$) in Fig. S2. If there is a signal at 166.18 Th ($C_{12}H_{22}^+$), its isotopologue, $^{13}CC_{11}H_{22}^+$ (166.175 Th), can be interfered on the signals of $C_{12}H_{23}^+$ (168.18 Th). Did the authors check such the interference of isotopologues on ion signals of C10-C20 cycloalkenes obtained in the filed measurements and in the chassis dynamometer study?

(3) Page 7, Line 169−Page 8, Line 192: I would like to see each mass spectrum of C10-C20 cycloalkanes (for example, n-alkylcyclohexane) and fractions of (m-1) for the cycloalkanes, like n-alkanes in Wang et al. (2020). According to Wang et al, (2020), it was reported that the fraction was relatively small for C8-C11 n-alkanes. I wonder if octylcyclohexane (C14), for example, produce strong fragment ions, leading to the interference on smaller cycloalkanes or not.

(4) Page 8, Line 193−Page 8, Line 209: Since 2-alkenes produce $C_nH_{2n}^+$ in addition to $C_nH_{2n-1}^+$ by NO$^+$ ionization (Diskin et al., 2002), check ion signals of $C_nH_{2n}^+$ in the filed

measurements and in the chassis dynamometer study. If there are signals, the authors should evaluate the interference.

**References:**

Diskin, A. M., Wang, T. S., Smith, D., and Španěl, P.: A selected ion flow tube (SIFT), study of the reactions of $H_3O^+$, $NO^+$ and $O_2^+$ ions with a series of alkenes; in support of SIFT-MS, International Journal of Mass Spectrometry, 218, 87−101, 2002.

Wang, C. M., Yuan, B., Wu, C. H., Wang, S. H., Qi, J. P., Wang, B. L., Wang, Z. L., Hu, W. W., Chen, W., Ye, C. S., Wang, W. J., Sun, Y. L., Wang, C., Huang, S., Song, W., Wang, X. M., Yang, S. X., Zhang, S. Y., Xu, W. Y., Ma, N., Zhang, Z. Y., Jiang, B., Su, H., Cheng, Y. F., Wang, X. M., and Shao, M.: Measurements of higher alkanes using $NO^+$ chemical ionization in PTR-ToF-MS: important contributions of higher alkanes to secondary organic aerosols in China, Atmospheric Chemistry and Physics, 20, 14123−14138, 2020.

---

## Referee Comment (RC2)

Chen et al. report the online measurement of cycloalkanes using the PTRMS instrument equipped with the NO chemical ionization scheme. The authors utilized a selection of authentic cyclohexane standards to showcase the high sensitivity, long term stability, low humidity dependence, and low detection limit of this method for the real time characterization of cyclohexane. With this method, C10-C20 cycloalkanes were measured in the ambient air at an urban site of China as well as from vehicular emissions in a chassis dynamometer experiment. The obtained concentrations of cycloalkanes can be as much as over half of the levels of the corresponding linear/branched alkanes, suggesting an appreciable amount of this group of compounds in the urban air. Overall, the authors have demonstrated very carefully the utilization of NO+PTRMS for the rigorous quantification of cycloalkanes in the field and laboratory. Here are a few thoughts and suggestions I would like the author to consider prior to publication on AMT.

Page 2, line 25: change 'undergoes' to 'undergo'.

Line 28 (and throughout the main text): 'as the result' should be 'as a result'. Line 35: change 'demonstrates' to 'demonstrate'. Line 38: add 'the' before 'importance'.

Page 9, line 221-225: Was there any measurement of cycloalkanes by PTRMS with the use of the NO+ ionization scheme? If so, could the authors compare their detection limits and sensitivities with those reports? If not, I am just curious why there was no attempt of using NO+PTRMS to detect cycloalkanes. After all, this ionization scheme has been out there for a while.

Page 9, line 226: Are the authors expecting a significant impact of water vapor on the instrument sensitivity? Unlike the proton transfer reaction, water cluster ion formation is not supposed to be a big issue on the ionization efficiency, right?

Page 10, line 254: For bicyclic alkanes, compounds such as unsaturated carbonyls share the identical molecular formula at the unit mass resolution. Could the TOF-MS provide a decent separation of these bicyclic alkanes from other potential interferences?

Page 10, line 255: What is the typical fraction of cycloalkanes in the overall carbon mass of organic compounds detected by NO+PTRMS? Is there any reason for smaller cycloalkanes (e.g., C6-C10) being excluded from the measurements and discussions?

Page 11, line 276-281: Again, I would like to see more discussions on the selectivity of NO+PTRMS towards cycloalkanes. Are there any other compounds that have been routinely measured by this NO+ ionization scheme? If so, how high are the signals of cycloalkanes compared with those compounds?

Page 12, line 316: please specify the technique used in these earlier studies.

Page 25: It seems like the PTR sensitivities to all these cycloalkanes are pretty close. I was wondering if the obtained average sensitivity can be used to other compounds that are detected by this NO+ scheme. Do the authors have any idea what instrument parameters or compound properties may affect the sensitivity?

Page 34: Are the authors expecting a large loss of these big alkane molecules to the sampling line?

---

## Author Response (AR1)

**Reviewer #1**

*General comments:*

*This paper describes a new technique of online measurements of cycloalkanes in the atmosphere by means of NO+ chemical ionization mass spectrometry. The authors used major product ions of $C_nH_{2n-1}+$ and $C_nH_{2n-3}+$ from NO+ chemical ionization of cyclic and bicyclic alkanes, respectively, for the measurement. After characterizing this technique in the laboratory, this technique was applied for field measurements at an urban site in southern China and a chassis dynamometer study regarding vehicle emissions. I think that this paper is generally well-written. However, I can't avoid a sense of apprehension because the authors showed the results of C10-C20 cyclic and bicyclic alkanes based on mass spectra of only five C7 and C12 cycloalkanes. The authors should show, for example, each mass spectrum of C10-C20 cycloalkanes (e. g., n-alkylcyclohexane) and discuss about interferences of other species on the ion signals of the cycloalkanes before applying the technique to the filed measurements and the chassis dynamometer study. I think that the characterizations shown in this paper are not enough. In addition, I am not sure that there is no interference on the data obtained in the filed measurements and in the chassis dynamometer study. Therefore, I recommend this paper to be revised referring to my specific comments listed below.*

Reply: We would like to thank the reviewer for the insightful comments, which helped us tremendously in improving the quality of our work.

In this study, we demonstrate that cyclic alkanes could be measured using the $NO^+$ chemical ionization via hydride ion transfer leading to M-H product ions. The major M-H ions are determined using commercially available chemicals including $C_7$, $C_{12}$, and $C_{15}$ cyclic alkanes and $C_{12}$ bicyclic alkanes (Table 1). These results are also complemented by the five different alkyl-cyclohexanes ($C_{10}$-$C_{14}$) in the cylinder gas standard (Table S1). It should be noted that these compounds are the only commercially available chemicals we can obtain. Though the number is limited, these compounds cover different types of cyclic alkanes, including alkyl-substituted cyclohexanes, cyclic compounds with more than 6 atom rings and also bicyclic compound. For example, we dedicatedly select all of the three types of cyclic alkanes with 12 carbons. Based on the

determined mass spectra of these cyclic alkanes, we can observe that the major product ions for all of these cyclic alkanes are M-H ions. As the result, we are convinced that other cyclic alkanes should follow the same reaction pathway with $NO^+$ yielding major product ions of M-H ions. To make this clear, we added the ionization results of $C_{15}$ cycloalkanes (cyclopentadecane and nonylcyclohexane) and $C_{10}$-$C_{14}$ alkyl-cyclohexanes in the revised manuscript, as shown in Fig. S5 and Fig. S6.

Fig. S5 shows mass spectra within the relevant range ($m/z$ $90^+$ to $240^+$ Th) for $C_{15}$ cycloalkanes. Similar with $C_7$ and $C_{12}$ cycloalkanes, the product ions generated by cyclopentadecane and nonylcyclohexane ($C_{15}H_{30}$) under $NO^+$ ionization mainly appear at $m/z$ 209 Th, corresponding to $C_{15}H_{29}^+$, but with more fragmentation ions. These results further verify that reactions of cycloalkanes with $NO^+$ ions follow the hydride ion transfer pathway to yield M-H product ions. Fig. S6 shows mass spectra within the relevant range ($m/z$ $130^+$ to $200^+$ Th) for $C_{10}$-$C_{14}$ alkyl-cyclohexanes during the calibration experiments. As shown in figure, five types of cycloalkanes produced regular mass spectrum peaks under $NO^+$ ionization, which appeared at $m/z$ 139 Th, $m/z$ 153 Th, $m/z$ 167 Th, $m/z$ 181 Th, and $m/z$ 195 Th, corresponding to $C_{10}H_{19}^+$, $C_{11}H_{21}^+$, $C_{12}H_{23}^+$, $C_{13}H_{25}^+$, and $C_{14}H_{27}^+$, respectively.

As for impurities, isotope, and other potential interference, we re-checked the data and evaluated these interferences. Please find the response to individual comments from the reviewer below.

The sentence in the Section 3.1 (line 195-198) is modified to:

**The product ions generated by cyclopentadecane and nonylcyclohexane ($C_{15}H_{30}$) mainly appear at $m/z$ 209 Th, corresponding to $C_{15}H_{29}^+$, with slightly more fragmentation than $C_{12}$ cyclic alkanes (Fig. S5).**

[Figure]

**Figure S5. Mass spectra of product ions from cyclopentadecane (a), and nonylcyclohexane (b) in NO⁺ PTR-ToF-MS. The major product ions are shown in red, and the fragments are shown in blue.**

The sentence in the Section 3.1 (line 200-202) is modified to:

**For instance, the mass spectra for $C_{10}$-$C_{14}$ alkyl-cyclohexanes during the calibration experiments are shown in Fig. S6, with the same $C_nH_{2n-1}^+$ as the major product ions.**

[Figure]

**Figure S6. Mass spectra of product ions from $C_{10}$-$C_{14}$ alkyl-cyclohexanes in NO⁺ PTR-ToF-MS during the calibration experiment.**

*Specific comments:*

*(1) Page 5, Line 123: I would like to know ion intensities of $O_{2+}$ and $NO_{2+}$ relative to that of $NO_+$ in a flow of zero (VOC-free) -air. I guess that the $O_{2+}$ was consumed by*

*reactions of VOCs in a sample air. The authors should evaluate the interference of the O$_{2+}$ reactions on the signals of cycloalkanes.*

Reply: We thank the reviewer for the comment. As NO$^+$ ions are used as parent ions in PTR-ToF-MS, the impurities such as O$_2^+$ and NO$_2^+$ do exist and may affect the measurements. Here, we present the intensities of NO$^+$ ions and other impurities including O$_2^+$, NO$_2^+$, and H$_3$O$^+$ during the measurements of urban air and vehicular emissions (Fig. S2). As shown in the two figures, the abundances of O$_2^+$, NO$_2^+$, and H$_3$O$^+$ ions are significantly lower than the NO$^+$ ions. We also calculated the ratio of O$_2^+$ to NO$^+$, and found that the ratio was below 5% during the measurements of urban air expect for the period from 26 October to 2 November, 2018 (7-10%), while the O$_2^+$/ NO$^+$ ratio was lower during the measurements of vehicular emissions, which is generally below 2%. Thus, the impurities inducing little interference to cycloalkanes detection.

The sentence in the Section 2.1 (line 128-134) is modified to:

**The intensities of primary ions NO$^+$ and impurities including O$_2^+$, NO$_2^+$, and H$_3$O$^+$ and the ratio of O$_2^+$ to NO$^+$ during the measurements of urban air and vehicular emissions are shown in Fig. S2. The abundances of O$_2^+$, NO$_2^+$, and H$_3$O$^+$ are significantly lower than NO$^+$ ions and the ratio of O$_2^+$ to NO$^+$ is basically below 5% during the measurements of urban air expect for the period from 26 October to 2 November, 2018 (7-10%), while the ratio of O$_2^+$/NO$^+$ is basically below 2% during the measurements of vehicular emissions.**

[Figure]

**Figure S2. Time series of NO$^+$, O$_2^+$, NO$_2^+$, and H$_3$O$^+$ during the measurements of urban air (a), and vehicular emissions (b-c).**

*(2) Page 6, Line 125: I would like to know ion signals at 166.18 Th (C$_{12}$H$_{22+}$) in Fig. S2. If there is a signal at 166.18 Th (C$_{12}$H$_{22+}$), its isotopologue, $_{13}$CC$_{11}$H$_{22+}$ (166.175 Th), can be interfered on the signals of C$_{12}$H$_{23+}$ (168.18 Th). Did the authors check such the interference of isotopologues on ion signals of C10-C20 cycloalkenes obtained in the filed measurements and in the chassis dynamometer study?*

Reply: We thank the reviewer for the comment. We re-examined the high-resolution peak fitting process for the data of urban air and the chassis dynamometer study, confirming that such isotopes would not interfere with the measurements of cycloalkanes in NO$^+$ PTR-ToF-MS.

During the high-resolution peak fitting, the signals contributed by ion isotopes have been calculated according to the proportional relationship between the ions and their isotopes in atmosphere (Stark et al., 2015; Timonen et al., 2016). As the result, the signals of cycloalkanes used for quantification is basically not affected by the contributions of isotopes from other ions. We added the isotopes corresponding to the identified mass spectrum peaks in Fig. S3. We also show the normalized signals of $C_{12}H_{23}^{+}$ and $^{13}CC_{11}H_{22}^{+}$ during the measurements of urban air in Fig. R1. The intensities of $^{13}CC_{11}H_{22}^{+}$ are significantly lower than $C_{12}H_{23}^{+}$, which further proved that the interference of isotopes to the measurements of cycloalkanes are little.

[Figure]

Figure. R1 Time series of $C_{12}H_{23}^{+}$ and $^{13}CC_{11}H_{22}^{+}$ during the measurement of urban air.

The sentence in the Section 2.1 (line 136-140) is modified to:

**The signal of cycloalkanes used for quantification has been subtracted from the contribution of isotopes from other ions and other species such as unsaturated aldehydes that share the identical formula at the unit mass resolution (UMR) with cycloalkanes during the high-resolution peak fitting process.**

[Figure]

**Figure S3. High-resolution peak fitting to the averaged mass spectra on a typical day (6 October 2018) for *m/z* 167 to individual ion peaks of $C_{12}$ cycloalkanes ($C_{12}H_{23}^+$), other isomeric ions ($C_8H_9ONO^+$, $C_{10}H_{14}O_2H^+$, and $C_{11}H_{19}O^+$), and isotopes of other ion ($^{13}CC_9H_{16}NO^+$ and $^{13}CC_{11}H_{22}^+$) detected from $NO^+$ PTR-ToF-MS.**

*(3) Page 7, Line 169−Page 8, Line 192: I would like to see each mass spectrum of C10-C20 cycloalkanes (for example, n-alkylcyclohexane) and fractions of (m-1) for the cycloalkanes, like n-alkanes in Wang et al. (2020). According to Wang et al, (2020), it was reported that the fraction was relatively small for C8-C11 n-alkanes. I wonder if octylcyclohexane (C14), for example, produce strong fragment ions, leading to the interference on smaller cycloalkanes or not.*

Reply: We thank the reviewer for the comment. In this study, we demonstrate that cyclic alkanes could be measured using the $NO^+$ chemical ionization via hydride ion transfer leading to M-H product ions. The major M-H ions are determined using commercially available chemicals including $C_7$, $C_{12}$, and $C_{15}$ cyclic alkanes and $C_{12}$ bicyclic alkanes (Table 1). These results are also complemented by the five different

alkyl-cyclohexanes ($C_{10}$-$C_{14}$) in the cylinder gas standard (Table S1). It should be noted that these compounds are the only commercially available chemicals we can obtain. Though the number is limited, these compounds cover different structures of cyclic alkanes, including alkyl-substituted cyclohexanes, cyclic compounds with more than 6 atom rings and also bicyclic compound. For example, we dedicatedly select all of the three types of cyclic alkanes with 12 carbons. Based on the determined mass spectra of these cyclic alkanes, we can observe that the major product ions for all of these cyclic alkanes are M-H ions. As the result, we are convinced that other cyclic alkanes should follow the same reaction pathway with $NO^+$ yielding major product ions of M-H ions.

We summarized the fractions of M-H ions produced by high-purity cycloalkanes species including $C_7$, $C_{12}$, and $C_{15}$ cyclic alkanes and $C_{12}$ bicyclic alkanes under the $NO^+$ ionization, as shown in Fig. S7. We observe that M-H ions account for ~100% of total ion signals for $C_7$ cyclic alkanes and lower but comparable fractions (74-82%) for $C_{12}$ and $C_{15}$ cyclic alkanes.

The sentences in the Section 3.1 (line 204-210) are modified to:

**The fractions of M-H ions generated by high-purity cycloalkanes species including $C_7$, $C_{12}$, and $C_{15}$ cyclic alkanes and $C_{12}$ bicyclic alkanes are summarized in Fig. S7. We observe that M-H ions account for ~100% of total ion signals for C7 cyclic alkanes and lower but comparable fractions (74-82%) for C12 and C15 cyclic alkanes. These results verify that reactions of cyclic and bicyclic alkanes with $NO^+$ ions follow the hydride ion transfer pathway to yield $C_nH_{2n-1}^+$ and $C_nH_{2n-3}^+$ product ions, respectively.**

[Figure]

**Figure S7. The fractions of product ions (M-H) from hydride abstraction of $C_7$, $C_{12}$, and $C_{15}$ cyclic alkanes and $C_{12}$ bicyclic alkanes in $NO^+$ PTR-ToF-MS.**

*(4) Page 8, Line 193−Page 8, Line 209: Since 2-alkenes produce $C_nH_{2n^+}$ in addition to $C_nH_{2n-1+}$ by $NO_+$ ionization (Diskin et al., 2002), check ion signals of $C_nH_{2n+}$ in the filed measurements and in the chassis dynamometer study. If there are signals, the authors should evaluate the interference.*

Reply: We thank the reviewer for the comment. We calculated and compared the signals of $C_nH_{2n}^+$ and $C_nH_{2n-1}^+$ in laboratory calibration experiments, urban air measurements and chassis dynamometer study (Fig. S8). The ratios of $C_nH_{2n}^+$ to $C_nH_{2n-1}^+$ signals of pure cyclic alkanes during the laboratory experiments maintained at 2-6%, similar to the ratios measured in filed measurements (3-7%). The ratios of $C_nH_{2n}^+$ to $C_nH_{2n-1}^+$ signals for vehicular emissions maintained at 6-16% for $C_{10}$-$C_{14}$ ions, which is slightly higher than those determined from cyclic alkanes, whereas the ratios for $C_{15}$-$C_{20}$ ions are comparable with pure cyclic alkanes (4-8%). Based on the results in Diskin et al. (2002), 2-alkenes produce 40-45% of $C_nH_{2n-1}^+$ and 55-60% of $C_nH_{2n}^+$ ions from $NO^+$ ionization. If we attribute all of the differences to potential interferences from 2-alkenes and assume the same quantities of $C_nH_{2n-1}^+$ and $C_nH_{2n}^+$ ions from $NO^+$ ionization from 2-alkenes, the interferences from alkenes should be in the range of 1-2% for urban air measurements and 2-12% for measurements of vehicular emissions. It should be noted that these number are upper limits, as 1-alkene may dominate the

$C_nH_{2n}^+$ ions and the signals from 2-alkenes may be significantly lower. This analysis further confirms that the interferences from alkenes to cyclic alkane measurements are minor.

The sentences in the Section 3.1 (line 233-245) are modified to:

**We further compare the signals of $C_nH_{2n}^+$ and $C_nH_{2n-1}^+$ from calibration experiments, urban air measurements and chassis dynamometer study (Fig. S8). The typical ratios of $C_nH_{2n}^+/C_nH_{2n-1}^+$ for cyclic alkanes are in the range of 2-6%, with similar ratios determined from urban air measurements (3-7%). The ratios of $C_nH_{2n}^+$ to $C_nH_{2n-1}^+$ from vehicular emissions maintained at 6-16% for $C_{10}$-$C_{14}$ ions, which is a little bit higher than those determined from cyclic alkanes, while the ratios of $C_{15}$-$C_{20}$ ions are comparable with pure cyclic alkanes (4-8%). Even though all of these differences are attributed to potential interferences from 2-alkenes and assume the same quantity of $C_nH_{2n-1}^+$ and $C_nH_{2n}^+$ ions from $NO^+$ ionization from 2-alkenes, the upper limits of the interferences from alkenes should be in the range of 1-2% for urban air measurements and 2-12% for measurements of vehicular emissions. Therefore, we conclude that the interferences from alkenes to cyclic alkanes measurements of cycloalkanes in most environments are minor.**

[Figure]

**Figure S8. The ratio of $C_nH_{2n}^+$ to $C_nH_{2n-1}^+$ from cycloalkanes (red), filed**

**measurement (blue) and vehicular emissions (green) measured by NO$^+$ PTR-ToF-MS.**

**Reviewer #2**

*Overview*

*Chen et al. report the online measurement of cycloalkanes using the PTRMS instrument equipped with the NO chemical ionization scheme. The authors utilized a selection of authentic cyclohexane standards to showcase the high sensitivity, long term stability, low humidity dependence, and low detection limit of this method for the real time characterization of cyclohexane. With this method, C10-C20 cycloalkanes were measured in the ambient air at an urban site of China as well as from vehicular emissions in a chassis dynamometer experiment. The obtained concentrations of cycloalkanes can be as much as over half of the levels of the corresponding linear/branched alkanes, suggesting an appreciable amount of this group of compounds in the urban air. Overall, the authors have demonstrated very carefully the utilization of NO+PTRMS for the rigorous quantification of cycloalkanes in the field and laboratory. Here are a few thoughts and suggestions I would like the author to consider prior to publication on AMT.*

Reply: We would like to thank the reviewer for the insightful comments, which helped us tremendously in improving the quality of our work. Please find the response to individual comments below.

*Specific Comments*

*1. Page 2, line 25: change 'undergoes' to 'undergo'.*

*Line 28 (and throughout the main text): 'as the result' should be 'as a result'.*

*Line 35: change 'demonstrates' to 'demonstrate'.*

*Line 38: add 'the' before 'importance'.*

Reply: We thank the reviewer for the comments. We corrected all these comments and checked the grammar throughout the manuscript.

*2. Page 9, line 221-225: Was there any measurement of cycloalkanes by PTRMS with the use of the NO+ ionization scheme? If so, could the authors compare their detection limits and sensitivities with those reports? If not, I am just curious why there was no attempt of using NO+PTRMS to detect cycloalkanes. After all, this ionization scheme*

*has been out there for a while.*

Reply: We thank the reviewer for the comment. When preparing this manuscript, we conducted a survey on the current literature, and found no relevant reports on the measurements of cycloalkanes in the atmosphere and emission sources by PTR-MS with $NO^+$ ionization, but there is a few laboratory studies have confirmed the ion chemistry (Koss et al., 2016) and use this technique to detect cycloalkanes in a smog chamber study (Wang et al., 2022a). We have included the reference of this literature in Section 1 (line 102-103).

*3. Page 9, line 226: Are the authors expecting a significant impact of water vapor on the instrument sensitivity? Unlike the proton transfer reaction, water cluster ion formation is not supposed to be a big issue on the ionization efficiency, right?*

Reply: We thank the reviewer for the comment. Previously, it was shown that response factors of higher acyclic alkanes in $NO^+$ PTR-ToF-MS are slightly affected by air humidity, and the degree of influence is related to carbon number (Wang et al., 2020a). Therefore, we also evaluate the influence of humidity on sensitivities of cycloalkanes in $NO^+$ PTR-ToF-MS using a custom-built humidity delivery system (Fig. S4), and the results are applied to explore the relationship between sensitivities of cycloalkanes and humidity. Figure R1 shows that normalized signals of $C_{10}$, $C_{11}$, $C_{13}$, and $C_{14}$ alkyl-cyclohexanes in the cylinder gas standard (Table S1) relative to dry conditions as a function of different humidity. The relative signals of the explored cycloalkanes show minor decrease (<10%) at the highest humidity (~82% RH at 25°C) compared to dry condition, and the observed changes for cycloalkanes with different carbon number are similar, suggesting little influence of humidity on measurements of cycloalkanes. The humidity-dependence curves determined in Figure R1 are used to corrected variations of ambient humidity in the atmosphere. The dependence of cyclic alkanes on humidity has been shown in Figure 4b and described in Section 3.2 in the original manuscript.

[Figure]

Figure R1. Humidity dependence of 1,1,3,5-teramethylcyclohexane **(a)**, pentylcyclohexane **(b)**, heptylcyclohexane **(c)**, and octylcyclohexane **(d)**.

*4. Page 10, line 254: For bicyclic alkanes, compounds such as unsaturated carbonyls share the identical molecular formula at the unit mass resolution. Could the TOF-MS provide a decent separation of these bicyclic alkanes from other potential interferences?*

Reply: We thank the reviewer for the comment. For those carbonyl compounds, it has been confirmed ketones will undergo association reactions with $NO^+$, while aldehydes are ionized with $NO^+$ via hydride ion transfer (Koss et al., 2016; Wang et al., 2020b). Therefore, the unsaturated aldehydes are more likely share the identical molecular formula at the unit mass resolution with bicyclic alkanes after $NO^+$ ionization. Figure R2 shows the high-resolution peak fitting to the averaged mass spectra on a typical day (6 October 2018) in urban air measurements for *m/z* 165, at which masses

produced by $C_{12}$ bicyclic alkanes ($C_{12}H_{21}^+$) and other isomeric ions ($C_8H_5O_4^+$, $C_9H_9O_3^+$, and $C_{10}H_{13}O_2^+$) detected from $NO^+$ PTR-ToF-MS. These results confirmed that high-resolution mass spectrometry peak fitting can provide good separation of these bicyclic alkanes from unsaturated aldehydes and other potential interferences.

[Figure]

Figure R2. High-resolution peak fitting to the averaged mass spectra on a typical day (6 October 2018) for *m/z* 165, at which masses produced by $C_{12}$ bicyclic alkanes ($C_{12}H_{21}^+$) and other isomeric ions ($C_8H_5O_4^+$, $C_9H_9O_3^+$, $C_{10}H_{13}O_2^+$, and $C_{11}H_{17}O^+$) detected from $NO^+$ PTR-ToF-MS.

*5. Page 10, line 255: What is the typical fraction of cycloalkanes in the overall carbon mass of organic compounds detected by NO+PTRMS? Is there any reason for smaller cycloalkanes (e.g., C6-C10) being excluded from the measurements and discussions?*

Reply: We thank the reviewer for the comment. Considering the complexity of the reaction pathways between organic compounds and $NO^+$ ions, we are not able to distinguish the total carbon mass of organic compounds detected by $NO^+$ PTR-ToF-MS. As for the smaller cycloalkanes (e.g., $C_6$-$C_{10}$), we are not able to achieve accurate quantitation in this study, as the customized cylinder gas standard used in calibration experiments only contain five different alkyl-cyclohexanes ($C_{10}$-$C_{14}$). In addition, as

shown in Fig. 2 and Fig. S7, larger cyclic alkanes show some fragment ions that are the same to the ions for C6-10 cyclic alkanes, especially $C_7H_{13}^+$ ion. As the result, measurements of $C_6$-$C_{10}$ may be slightly affected by fragmentation of the larger cycloalkanes, which warrants further investigation.

*6. Page 11, line 276-281: Again, I would like to see more discussions on the selectivity of NO+PTRMS towards cycloalkanes. Are there any other compounds that have been routinely measured by this NO+ ionization scheme? If so, how high are the signals of cycloalkanes compared with those compounds?*

Reply: We thank the reviewer for the comment. The time series of toluene and acetaldehyde shown in Fig. 7 were actually detected by $NO^+$ ionization by PTR-ToF-MS. Toluene and acetaldehyde are ionized with $NO^+$ via charge transfer and hydride ion transfer leading to major product ions of $C_7H_8^+$ and $C_2H_3O^+$, respectively. As shown in Figure R3, good agreement between PTR-ToF-MS with $H_3O^+$ and $NO^+$ chemistry was obtained for aromatics and oxygenated VOCs during the measurements of urban air, which has been reported in a recent study during the same campaign (Wang et al., 2020a). Therefore, we used toluene and acetaldehyde data from $NO^+$ measurement in vehicular emissions. The concentrations of cyclic and bicyclic alkanes are lower than acetaldehyde both in gasoline and diesel emissions, with ratios account for less than 1% of acetaldehyde, while the concentrations of cyclic and bicyclic alkanes are lower than toluene in gasoline emissions (with ratios account for 1-3% of toluene concentration) and about 5-10 times than toluene in diesel emissions.

[Figure]

Figure. R3 Comparisons of benzene, toluene, acetaldehyde, and pentanone measured by NO⁺ PTR-ToF-MS (red dots) and H₃O⁺ PTR-ToF-MS (blue dots) during the measurements of urban air.

The sentences in page 36 (line 728-733) are modified to:

**Figure 7. Concentrations of $C_{12}$ cyclic, bicyclic, and acyclic alkanes, acetaldehyde, toluene, and $CO_2$ for (a) a gasoline vehicle with emission standard of China III and (b) a diesel vehicle with emission standard of China IV. The gray shadows represent the speeds of the vehicles on the chassis dynamometer. The data of toluene and acetaldehyde were detected by NO⁺ PTR-ToF-MS.**

*7. Page 12, line 316: please specify the technique used in these earlier studies.*

Reply: We thank the reviewer for the comment. We summarized the techniques

used in these earlier studies and the results are shown in Table S2 in the revised manuscript.

The sentences in the Section 3.4 (line 354-358) are modified to:

**As there are only limited measurements of bicyclic alkanes in the literature, we compare concentration ratios of cyclic alkanes to acyclic alkanes with results in previous studies, mainly using gas chromatography techniques (GC-MS/FID and GC×GC). The details of the technique used in these earlier studies are summarized in Table S2.**

**Table S2. Detailed information of the measurement locations and techniques used for detection of cycloalkanes in this study and previous studies.**

| Measurement location | Measuring techniques | References |
|---|---|---|
| Guangzhou, China | NO$^+$ PTR-ToF-MS | This work |
| Algiers, Algeria | High-resolution gas chromatography-mass spectrometry (GC-MS) [a] | (Yassaa et al., 2001) |
| Los Angeles, USA | A two-channel in situ gas chromatography-mass spectrometry (GC-MS/FID) [a] | (de Gouw et al., 2017) |
| London, UK | Two-dimensional gas-chromatography time-of flight mass- spectrometry (TD-GC×GC ToF-MS) [c] | (Xu et al., 2020) |
| Northeastern Colorado, USA | GC-MS/FID [a] | (Gilman et al., 2013) |
| Lubricating oil | TD-GC×GC ToF-MS [b] | (Liang et al., 2018) |
| Diesel exhausts | TD-GC×GC ToF-MS [b] | (Alam et al., 2016) |
| Gasoline and diesel exhausts | GC-MS [c] | (Gentner et al., 2012) |
| Gasoline and diesel exhausts | NO$^+$ PTR-ToF-MS | This work |

**[a] The reported acyclic and cyclic alkanes were identified and quantified with gas standards**

**[b] The total ions signals of species is integrated into different regions (bin) according to the residence time of *n*-alkanes. The total ions signals of each bin were considered as the signals of acyclic and cyclic alkanes.**

**[c] The total ions signals of acyclic and cyclic alkanes were calculated by subtracted the signals of known compounds from similar chemical classes, and the remaining signals were considered to be the signals of acyclic and cyclic alkanes.**

*8. Page 25: It seems like the PTR sensitivities to all these cycloalkanes are pretty close. I was wondering if the obtained average sensitivity can be used to other compounds that are detected by this NO+ scheme. Do the authors have any idea what instrument parameters or compound properties may affect the sensitivity?*

Reply: We thank the reviewer for the comment. Based on a series of laboratory experiments, we found that the calibrated cycloalkanes show similar sensitivities. Therefore, we use the average sensitivity of $C_{10}$-$C_{14}$ cycloalkanes to predict the sensitivities of cycloalkanes with higher carbon numbers ($C_{15}$-$C_{20}$).

However, we do not have much information on the parameters or compound properties that can affect sensitivities of cyclic alkanes at present. Per suggestion of reviewer #1, we checked the fragmentation of various cyclic alkanes, and we did not observe significant variations of fragmentation for cyclic alkanes with different carbon number, different sizes of the ring and the substituted alkyl group. This observation may explain similar sensitivities of cyclic alkanes. In contrast to cyclic alkanes, sensitivities of acyclic alkanes strongly correlate with the fragmentation degree of ions, with lower potential of fragmentation for acyclic alkanes with higher carbon number, as shown in Figure 4 of our previous study of NO$^{+}$ chemical ionization (Wang et al., 2020).

We added the description in the line 681-683: **The average sensitivity of $C_{10}$-$C_{14}$ cyclic alkanes was used to predict the concentrations of cyclic alkanes with higher carbon ($C_{15}$-$C_{20}$) and bicyclic alkanes ($C_{10}$-$C_{20}$).**

*9. Page 34: Are the authors expecting a large loss of these big alkane molecules to the sampling line?*

Reply: We thank the reviewer for the comment. The perfluoroalkoxy (PFA) Teflon tubing is used for inlets in the measurements of cycloalkanes for both urban air and vehicular emissions, but gas-wall partitioning can be important for intermediate and semi-volatility compounds (Pagonis et al., 2017). In this study, we use the delay time to determine response of cycloalkanes and evaluate the interference caused by gas-wall partitioning. The delay time of cycloalkanes are summarized in Fig. 5, which

ranging from a few seconds to a few minutes. These results suggest that measured variability of cycloalkanes with higher carbon number, especially for $C_{19}$-$C_{20}$ or above, may only be reliable for time scales longer than 10 min. However, as shown in Fig. S13, the larger cyclic and bicyclic alkanes ($C_{19}$-$C_{20}$) exhibit very similar diurnal variations to smaller cyclic and bicyclic alkanes ($< C_{19}$) during the measurements of urban air, implying that the gas-wall partitioning should not significantly affect temporal variations of larger cyclic and bicyclic alkanes reported in this study.

**Reviewer #3**

*Overview*

*The authors investigate the simultaneous detection of cycloalkanes and acyclic alkanes using a PTR-TOF-MS with NO+ ionization chemistry. The measurement technique was tested in a laboratory before applied in the field, here the city of Guangzhou, China and at a chassis dynamometer. The authors confirm that cyclic alkanes are ionized via hydride ion transfer while isomers of alkanes, alkenes, cluster with NO+. Using a gas standard containing cyclic and acyclic molecules their sensitivity was determined. Effects of humidity and tubing were also considered by the authors. Their motivation is the contribution of cyclic and bicyclic alkanes to the formation of SOA in urban environments or from vehicle exhaust. They also report different ratios for cyclic and acyclic alkanes in diesel and gasoline exhaust.*

*Further investigation of the application of PTR-TOF-MS with NO+ ionization chemistry is of great interest for the field if VOC detection. The results provide a new dataset for C10 to C20 alkanes under polluted conditions. I thus recommend that the work is published.*

Reply: We would like to thank the reviewer for the insightful comments, which helped us in improving the quality of our work. Please find the response to individual comments below.

*Specific comments:*

*1. Line 100: This sentence sounds as if cycloalkanes and acyclic alkanes have never been measured simultaneous in ambient air before, but Koss et al., 2016 did that to my knowledge.*

Reply: We thank the reviewer for the comment. We corrected this sentence in the Section 1 (line 100-103) as **"These evidences suggest that NO$^+$ ionization scheme could provide a possibility for measuring cycloalkanes along with acyclic alkanes, as demonstrated in two recent work (Koss et al., 2016; Wang et al., 2020a) "**.

*2. Line 118: The authors show one part of the ionization sequence, but the formation of NO$^+$ ions in the ion source is more complex. See Karl et al., 2012 (doi:10.5194/acp-12-*

*11877-2012).*

Reply: We thank the reviewer for the comment. We supplemented the ionization sequence of the formation of $NO^+$ ions in manuscript.

The sentences in the Section 2.1 (line 115-122) are modified to:

**In order to generate $NO^+$ ions, 5 sccm ultra-high-purity air ($O_2+N_2 \geq$ 99.999%) is directed into to the hollow cathode discharge area of ion source, $NO^+$ ions are produced by ionization as follows (Federer et al., 1985; Karl et al., 2012):**

$$N^+ + O_2 \rightarrow NO^+ + O \quad\quad\quad (a)$$

$$O^+ + N_2 \rightarrow NO^+ + N \quad\quad\quad (b)$$

$$N_2^+ + O_2 \rightarrow O_2^+ + N_2 \quad\quad\quad (c)$$

$$O_2^+ + NO \rightarrow NO^+ + O_2 \quad\quad\quad (d)$$

*3. Line 120: The authors state that impurities are minimum, but no values are given. Also it is known from Yuan et al., 2016 that primary ion signals as well as signals of the impurities can be influenced by the ion guide. This study used an instrument including an ion guide, thus it would be important to rule out artefacts arising from that.*

Reply: We thank the reviewer for the comment. When $NO^+$ ions are used as parent ions in PTR-ToF-MS, the impurities such as $O_2^+$ and $NO_2^+$ do exist and may affect the measurements. Here, we present the intensities of $NO^+$ ions and other impurities including $O_2^+$, $NO_2^+$, and $H_3O^+$ during the measurements of urban air and vehicular emissions (Fig. S2). As shown in the two figures, the abundances of $O_2^+$, $NO_2^+$, and $H_3O^+$ ions are significantly lower than the $NO^+$ ions. We also calculated the ratio of $O_2^+$ ions to $NO^+$ ions, and found that ratio was below 5% during the measurements of urban air expect for the period from 26 October to 2 November, 2018 (7-10%), while the $O_2^+/ NO^+$ ratio was lower during the measurements of vehicular emissions, which is generally below 2%. Thus, the impurities inducing little interference to cycloalkanes detection.

The sentence in the Section 2.1 (line 128-134) is modified to:

**The intensities of primary ions $NO^+$ and impurities including $O_2^+$, $NO_2^+$, and**

$H_3O^+$ and the ratio of $O_2^+$ to $NO^+$ during the measurements of urban air and vehicular emissions are shown in Fig. S2. The abundances of $O_2^+$, $NO_2^+$, and $H_3O^+$ are significantly lower than $NO^+$ ions and the ratio of $O_2^+$ to $NO^+$ is basically below 5% during the measurements of urban air expect for the period from 26 October to 2 November, 2018 (7-10%), while the ratio of $O_2^+/NO^+$ is basically below 2% during the measurements of vehicular emissions.

[Figure]

**Figure S2. Time series of $NO^+$, $O_2^+$, $NO_2^+$, and $H_3O^+$ during the measurements of urban air (a), and vehicular emissions (b-c).**

*4. Line 130: VOCs can also cluster with H3O+, I suggest writing: Compared to proton transfer reactions occurring mostly between H3O+ ions and VOCs species…*

Reply: We thank the reviewer for the comment. We corrected this sentence in the Section 2.1 (line 144-146) as **"Compared to proton transfer reactions occurring mostly between H₃O⁺ ions and VOCs species, NO⁺ ions show a variety of reaction pathways with VOCs, which can be roughly summarized as follow"**.

5. *Line 209: As far as I understood the interferences can still be up to 15 %. To my opinion this is worth mentioning like ( < 15 % ).*

Reply: We thank the reviewer for the comment. We corrected this sentence in the Section 3.1 (line 230-233) as **"However, concentrations of 1-alkenes and trans-2-alkenes in the atmosphere are usually significantly lower than cycloalkanes (about 25% and <15%, respectively)"**.

6. *Line 211: The authors mention before that the calibration experiments were done with a gas standard containing compounds listed in Table S1, but the information would be very helpful in this chapter as well.*

Reply: We thank the reviewer for the comment. We corrected this sentence in the Section 3.2 (line 247-249) as **"The calibration experiments of cycloalkanes (see details of gas standard in Table S1) are carried out in both dry conditions (<1% RH) and humidified conditions (Fig. S9)"**.

7. *Line 238: The authors show how vehicular emissions drop to 10 %, but how did they measure that technically. Was the inlet brought close to the cars exhaust? How was the switching from detecting exhaust to clean air done? I imagine they used a dilutive flow of synthetic air, but this is not described in the manuscript.*

Reply: We thank the reviewer for the comment. The delay time of cycloalkanes in vehicular emissions was calculated based on the results of chassis dynamometer study, so the sampling methods are same as mentioned in Li et al. (2021) and Wang et al. (2022b). A custom-built sampling and dilution system for vehicles combining online and offline sampling techniques was used during this campaign. The vehicles were pushed onto a chassis dynamometer and driven through short transient driving cycle for

3~5 times. During the test, vehicle exhaust is diluted by a factor of 10-100 by zero air using the custom-built dilution system. In other words, the drop of signals of cyclic alkanes for vehicular emissions are as the result of switching from diluted vehicular exhausts to zero air.

The sentence in the Section 3.2 (line 271-274) is modified to:

**For the species not in the gas standard, we also take advantage of vehicular emissions measurements associated with high concentrations of cycloalkanes, and the sampling methods are same as mentioned in Li et al. (2021) and Wang et al. (2022b).**

*8. Line 249: Did the authors calculate average sensitivities for cyclic and bicyclic alkanes separately? The fragmentation seems to be different.*

Reply: We thank the reviewer for the comment. We did not calculate the average sensitivities for cyclic and bicyclic alkane separately, as bicyclic alkanes are not contained in the gas standard. For bicyclic alkanes, we also use the average sensitivity of C10-C14 monocyclic alkanes.

The fragmentation of various compounds is shown in Figure S7 in the revised manuscript. From Figure S7, the fractions of product ions (M-H) from hydride abstraction in the total product ions for bicyclohexane is 71%, which is only slightly lower than monocyclic alkanes with the same carbon number (cyclodecane~77% and hexylcyclohexane~75%). Based on this only information for the fragmentation of bicyclic alkanes, we tentatively assume bicyclic alkanes fragment similar with other compounds and the average sensitivity of C10-C14 monocyclic alkanes.

We added the description in the line 681-683: **The average sensitivity of $C_{10}$-$C_{14}$ cyclic alkanes was used to predict the concentrations of cyclic alkanes with higher carbon ($C_{15}$-$C_{20}$) and bicyclic alkanes ($C_{10}$-$C_{20}$).**

[Figure]

**Figure S7. The fractions of product ions (M-H) from hydride abstraction of C$_7$, C$_{12}$, and C$_{15}$ cyclic alkanes and C$_{12}$ bicyclic alkanes in NO$^+$ PTR-ToF-MS.**

*9. Line 269: Since there are also cases where cyclic alkanes are more abundant than acyclic alkanes I suggest writing: ... suggestion they predominantly came from same emission sources.*

Reply: We thank the reviewer for the comment. We corrected this sentence in the Section 3.3 (line 305-308) as **"Based on both time series and correlation analysis (Fig. 6c), cyclic and bicyclic alkanes showed strong correlation with acyclic alkanes, suggesting they predominantly came from same emission sources"**.

*10. Line 289: The authors report completely different emission pattern from diesel vehicles compared to gasoline. Has this been detected before for other compounds? Is there a known explanation for the difference?*

Reply: We thank the reviewer for the comment. In addition to cycloalkanes and alkanes, other compounds such as aromatics and oxygenated VOCs also presents similar different emission pattern between gasoline and diesel vehicles, which has been reported in a recent study during the same campaign (Wang et al., 2022b). The different emission patterns from diesel vehicles compared to gasoline vehicles for cycloalkanes and alkanes can be explained by chemical compositions of gasoline and diesel fuels are different. The gasoline fuel used in China is mainly comprised of C$_4$-C$_7$ hydrocarbons,

including alkanes (55%-62%), alkenes (12%-17%), aromatics (27%-32%), and methyl tert-butyl ether (MTBE, 1%-4%) (Huang et al., 2022; Qi et al., 2021; Sun et al., 2021; Tang et al., 2015), while heavy hydrocarbons, mainly $C_8$-$C_{10}$ alkanes and aromatics account for major fractions in diesel fuel, including alkanes (70%-79%), alkenes (1%-7%), and aromatics (21%-25%) (Yue et al., 2015).

The sentences in the Section 3.3 (line 329-332) are modified to:

**In addition to cycloalkanes and alkanes, other compounds including aromatics and oxygenated VOCs also demonstrate large differences between gasoline and diesel vehicles, which were mainly attributed to different chemical compositions of gasoline and diesel fuels (Wang et al., 2022b).**

*11.Line 315: For some alkanes ($C_{15}$, $C_{16}$, $C_{18}$) the ratio observed in London is much larger than detected in this work. This is not similar. I would appreciate a more detailed discussion at this point.*

Reply: We thank the reviewer for the comment. We re-checked measurement results of cyclic and acyclic alkanes from Xu et al. (2020), and found an error in processing this dataset from London. We corrected the ratios obtained in London and modified Fig. 9. The ratios obtained in London are higher than the ratios obtained in the urban region of Guangzhou, but the ratios in London are similar to diesel exhausts in our work for $C_{13}$-$C_{18}$ range. These results are likely due to the measurement location in London is proximity to a main road, where cyclic and acyclic alkanes may be dominated by traffic emissions with high fractions of diesel vehicles in the fleet. Although some variations observed in different urban environments, nevertheless, these ratios are broadly within the range between gasoline and diesel emissions.

The sentences in the Section 3.4 (line 358-367) and Fig. 9 are modified to:

**As shown in the Fig. 9, the ratios obtained in the urban region of Guangzhou in this work (0.2-0.4) are similar to other measurements in urban area, including Algiers, Algeria (Yassaa et al., 2001). The ratios obtained in London, UK (Xu et al., 2020) are higher than the ratios obtained in Guangzhou, but similar to the diesel exhaust in our work for $C_{13}$-$C_{18}$ range. These results are likely due to the**

measurement location in London is proximity to a main road, cyclic and acyclic alkanes may be dominated by traffic emissions with high fractions of diesel vehicles in the fleet. Although some variations observed in different urban environments, nevertheless, these ratios are broadly within the range between gasoline and diesel emissions.

[Figure]

Figure 9. The concentrations ratios of cyclic alkanes to acyclic alkanes for different carbon number. Measurements in various urban areas, including Guangzhou in China, London in UK (Xu et al., 2020), Los Angeles in US (de Gouw et al., 2017), Algiers in Algeria (Yassaa et al., 2001), and an oil and gas region in Colorado of US (Gilman et al., 2013) are also shown for comparison. Emission sources, including vehicle exhausts (Alam et al., 2016; Gentner et al., 2012) and lubricating oils (Liang et al., 2018) are also included.

*12. Figure 8: Here it would be very helpful to see error bars. AS the authors present averaged values the variability is important to proof significance, especially for the comparison of the ratios.*

Reply: We thank the reviewer for the comment. We have added error bars on the corresponding figure. Figure 8 (line 728-733) is modified to:

[Figure]

**Figure 8. Mean concentrations of cyclic and bicyclic alkanes and alkanes (branched + linear) with different carbon numbers measured by NO+ PTR-ToF-**

**MS in the urban air (a) and diesel emissions (b). The green and purple lines with circles represent the ratios of cyclic and bicyclic alkanes to acyclic alkanes under the same carbon numbers, respectively. Error bars represent standard deviations of the concentration for the acyclic, cyclic and bicyclic alkanes.**

*Technical corrections:*

*1. Line 31: For a better reading I recommend writing: Appling this method, cycloalkanes were successfully measured at an urban site in southern China and during a chassis dynamometer study for vehicular emissions.*

Reply: We thank the reviewer for the comment. We corrected this sentence in Abstract (line 30-32) as **"Appling this method, cycloalkanes were successfully measured at an urban site in southern China and during a chassis dynamometer study for vehicular emissions"**.

*2. Line 35: These results demonstrate that NO+ PTR-ToF-MS…*

Reply: We thank the reviewer for the comment. We corrected this sentence in Abstract (line 35-38) as **"These results demonstrate that $NO^+$ PTR-ToF-MS provides a new complementary approach for fast characterization of cycloalkanes in both ambient air and emission sources, which can be helpful to fill the gap in understanding the importance of cycloalkanes in the atmosphere"**.

*3. Line 45: Components and concentration levels of organic compounds largely affect atmospheric chemistry, …*

Reply: We thank the reviewer for the comment. We corrected this sentence in the Section 1 (line 45-48) as **"Components and concentration levels of organic compounds largely affect atmospheric chemistry, atmospheric oxidation capacity, and radiation balance (Monks et al., 2015; Wu et al., 2020), as well as human health (Xing et al., 2018) "**.

*4. Line 75: For a better understanding I recommend writing: Based on measurements*

*of gas chromatographic techniques, the signals of unspeciated cyclic compounds can be determined. This is done by subtracting the signal of speciated IVOC from the total signal for each retention time bin according to the series of n-alkanes.*

Reply: We thank the reviewer for the comment. We corrected this sentence in the Section 1 (line 75-78) as **"Based on measurements of gas chromatographic techniques, the signals of unspeciated cyclic compounds can be determined. This is done by subtracting the total signal for each retention time bin according to the series of n-alkanes (Zhao et al., 2014; Zhao et al., 2016) "**.

*5. Line 103: Typo: … ambient air and from emission sources…*

Reply: We thank the reviewer for the comment. We corrected this sentence in the Section 1 (line 104-105) as **"In this study, we discuss the potential of online measurements of cycloalkanes in ambient air and emission sources utilizing NO⁺ ionization in PTR-ToF-MS"**.

*6. Line 105: I suggest to write: The results of laboratory experiments to characterize product ions,…*

Reply: We thank the reviewer for the comment. We corrected this sentence in the Section 1 (line 105-107) as **"The results of laboratory experiments to characterize product ions, calibration, and response time will be shown"**.

*7. Line 113: mass resolution instead of mass resolving*

Reply: We thank the reviewer for the comment. We corrected this sentence in the Section 2.1 (line 112-115) as **"A commercially PTR-ToF-MS instrument (Ionicon Analytik, Austria) equipped with a quadrupole ion (Qi) guide for effective transfer of ions from drift tube to the time-of-flight mass spectrometer is used for this work (Sulzer et al., 2014), and the mass resolution approximately reach about 3000 m/Δm (Fig. S1)"**.

*8. Line 147: In this study, we investigate characteristic ions of cycloalkanes generated*

*by the NO+ ionization…*

Reply: We thank the reviewer for the comment. We corrected this sentence in the Section 2.2 (line 161-162) as **"In this study, we investigate characteristic ions of cycloalkanes generated by NO+ ionization from a series of species identification experiments".**

*9. Line 149: I suggest using species instead of chemicals.*

Reply: We thank the reviewer for the comment. We corrected this sentence in the Section 2.2 (line 162-164) as **"The information of cycloalkanes species used in these experiments is listed in Table 1".**

*10. Line 160: Typo: sensitivities*

Reply: We thank the reviewer for the comment. We corrected this sentence in the Section 2.2 (line 173-176) as **"Therefore, we evaluate the influence of humidity on sensitivities of cycloalkanes in NO+ PTR-ToF-MS using a custom-built humidity delivery system (Fig. S4), and the results are applied to explore the relationship between sensitivities of cycloalkanes and humidity".**

*11. Line 186: The sentence is confusing to me, I suggest writing: As mentioned above, thecharacteristic peaks of cycloalkanes under NO+ ionization are consistent with the ions that are received at the attempts to utilize H3O+ PTR-MS. For the latter method though sensitivities are reported to be lower.*

Reply: We thank the reviewer for the comment. We corrected the sentences in the Section 3.1 (line 210-213) as **"As mentioned above, the characteristic peaks of cycloalkanes under NO$^+$ ionization are consistent with the ions that are received at the attempts to utilize H$_3$O$^+$ PTR-MS. For the latter method though sensitivities are reported to be lower (Erickson et al., 2014; Gueneron et al., 2015; Warneke et al., 2014; Yuan et al., 2014)".**

*12. Line 193: The isomers… (without 'as')*

Reply: We thank the reviewer for the comment. We corrected this sentence in the Section 3.1 (line 218-219) as **"The isomers of cyclic alkanes, alkenes may interfere with measurements of cycloalkanes"**.

13. Line 202: I recommend adding: ...which are similar fragmentation ions from NO+ ionization of the two species and ...

Reply: We thank the reviewer for the comment. We corrected this sentence in the Section 3.1 (line 225-228) as **"Based on the mass spectra, alkenes produce the $C_nH_{2n-1}^+$ product ions at fractions of <5%, which are similar fragmentation ions from $NO^+$ ionization of the two species and other 1-alkenes determined from a selected ion flow tube mass spectrometer (SIFT-MS) (Diskin et al., 2002)".**

14. Line 212: Figure S4 is never mentioned in the manuscript.

Reply: We thank the reviewer for the comment. We have re-checked the figure name of the manuscript and found that it was wrongly marked, which has been modified in the corresponding position of this manuscript.

The sentence in the Section 3.2 (line 247-249) is modified to:

**The calibration experiments of cycloalkanes (see details of gas standard in Table S1) are carried out in both dry conditions (<1% RH) and humidified conditions (Fig. S9).**

15. Line 241: Typo: ...but relatively lower than determined for those acyclic alkanes.

Reply: We thank the reviewer for the comment. We corrected this sentence in the Section 3.2 (line 277-280) as **"The delay time of various cycloalkanes generally increases with the carbon numbers, ranging from a few seconds to a few minutes, but relatively lower than determined for those acyclic alkanes within C10-C15 range (Wang et al., 2020a) ".**

16. Line 307: Fig. 8b instead of Fig. 9

Reply: We thank the reviewer for the comment. We corrected this sentence in the

[revised manuscript text omitted]